# Hierarchical Object-Oriented POMDP Planning for Object Rearrangement

## Abstract

We present an online planning framework for solving multi-object rearrangement problems in partially observable, multi-room environments. Current object rearrangement solutions, primarily based on Reinforcement Learning or hand-coded planning methods, often lack adaptability to diverse challenges. To address this limitation, we introduce a novel Hierarchical Object-Oriented Partially Observed Markov Decision Process (HOO-POMDP) planning approach. This approach comprises of (a) an object-oriented POMDP planner generating sub-goals, (b) a set of low-level policies for sub-goal achievement, and (c) an abstraction system converting the continuous low-level world into a representation suitable for abstract planning. We evaluate our system on varying numbers of objects, rooms, and problem types in AI2-THOR simulated environments with promising results.

## 1 Introduction

Multi-object rearrangement with egocentric vision in realistic home environments is a fundamental challenge in embodied AI, encompassing complex tasks that require perception, planning, navigation, and manipulation. This problem becomes particularly demanding in multi-room settings with partial observability, where large parts of the environment are not visible at any given time. Such scenarios are ubiquitous in everyday life, from tidying up households to organizing groceries, making them critical for the development of next-generation home assistant robots.

Existing approaches to multi-object rearrangement typically fall into two categories: Reinforcement Learning (RL) methods and hand-coded planning systems. RL methods often struggle as the problem becomes increasingly complex and lengthy, making it difficult to scale to more challenging scenarios. To address this limitation, many researchers have adopted a modular approach, decomposing the task into a series of subtasks (Gu et al., 2022) such as manipulation skills, navigation skills, or exploration skills. These subtasks are then sequenced together in different ways to accomplish the overall goal. However, current modular approaches have their own limitations. Some pre-determine the sequence in which to apply skills, while others use greedy planners (Trabucco et al., 2022). This constrains their potential for full optimization in terms of determining the optimal order to interact with objects and handling new problems, such as when another object blocks the path to an object or location or if the goal location itself is obstructed. To overcome these challenges, a more general approach that incorporates high-level planning based on the current state would be beneficial. Such an approach would enable the system to handle novel problems without requiring extensive re-learning from scratch, thus increasing its adaptability and efficiency in diverse environments. Solving these new problems is particularly important in the context of household robots, where obstacles, blocked paths, and obstructed goals are common occurrences that a robust system must be able to handle effectively.

While significant progress has been made in rearrangement, the majority of current research focuses on single-room settings or assumes that a large number of objects are visible at the beginning of the task, either through a third-person bird's eye view (Ghosh et al., 2022) or a first-person view where most of the room is visible. However, as we move towards the more practical version of the problems, such as cleaning a house, the majority of the space and objects to be manipulated are not initially visible, and existing solutions begin to falter. This scenario of partial observability introduces several major challenges: 1) uncertainty over object locations, as the starting positions of objects are unknown; 2) execution efficiency of searching for objects while simultaneously moving

them to the correct goal locations; 3) scalability of planning over increasing numbers of objects and rooms; 4) extensibility to scenarios involving blocked goals or obstructed paths; and 5) graceful handling of object detection failures.

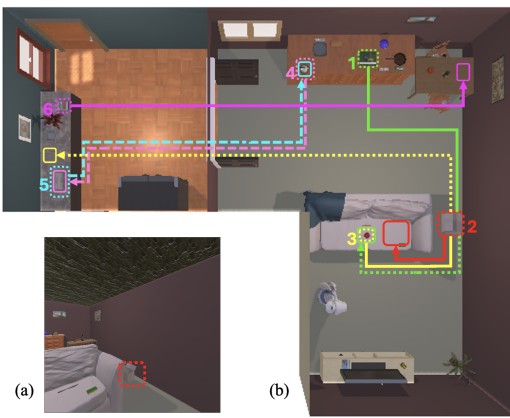

Figure 1: a) shows the agent's ego-centric view of the world at initialization. b) shows the top-down view of the environment before the start of the re-arrangement task. The dotted bounding boxes indicate the starting state of the object, and the solid bounding boxes indicate the object's goal state. In a) The lines indicate the path between the start and goal states. Object 1's path to goal is blocked by object 2, and its goal location is blocked by object 3. Object 3's path to its goal is also blocked by object 2. Hence, object 2 needs to be moved to its goal, then object 3 to its goal, and only then can object 1 be moved to its goal. Objects 4 and 5 are blocking each other's goals, and hence, one of them needs to be placed elsewhere, and then the swap can happen.

We employ a hierarchical approach to decompose the complex multi-object rearrangement problem into two distinct levels. This structure allows us to address object combinatorics at the high level while managing regional interactions at the low level. By separating these aspects, we effectively mitigate the challenges inherent to each. We use an object-oriented representation rather than low-level perceptual representations such as point clouds, which reduces the complexity of planning and is more natural in our setting. Our hierarchical Object-Oriented Partially Observed Markov Decision Process (HOO-POMDP) planning framework has the following components: 1) A perception module that detects objects and outputs object-oriented observations, 2) A belief update and state abstraction system, 3) A high-level planning module that plans the sequence of locations to be visited or objects to be transported, 4) A low-level navigation and manipulation module that plans a path to its next destination or control actions to pick and place objects.

In more detail, the agent maintains a factored belief state over object locations based on its prior beliefs and the output of its object detection module. An abstract object-oriented POMDP planner produces a high-level policy for the given goal. The first high-level action in the plan is treated as a (navigation or manipulation) sub-goal for the low-level planner, which produces a lower-level sequence of actions. The agent takes the first action in the lower-level plan, updates its belief state based on its new perception, and the cycle repeats.

The main contributions of our research are as follows:

- A modular planning system, which includes an object-oriented planner and a state abstraction module for object rearrangement in multi-room environments.
- A new dataset featuring blocked path problems and expanded room configurations alongside existing rearrangement challenges.
- An empirical evaluation of the system in the new dataset in AI2Thor under different conditions.

## 2 RELATED WORK

**Rearrangement:** Rearrangement is the problem of manipulating the placement of objects by picking, moving, and placing them according to a goal configuration. In this work, we are mainly concerned with the rearrangement of objects by mobile agents in simulated environments such as AI2Thor and Habitat (Kolve et al., 2017),(Szot et al., 2021) and (Gan et al., 2020). We hypothesize that successful rearrangement planning in simulators with noisy sensors and effectors will go a long way towards successful planning and execution in robotics.

There are many versions of the rearrangement problem in literature. In tabletop rearrangement, a robot hand with a fixed base moves objects around to achieve a certain configuration in a limited

space (Zhai et al., 2024; Huang et al., 2024). Many current approaches for rearrangement by mobile agents work by finding the misplaced objects and then use greedy planners to decide what order to move the objects in (Gadre et al., 2022; Trabucco et al., 2022; Sarch et al., 2022). This can lead to a high traversal cost since it is not explicitly optimized. The above works and others such as (Mirakhor et al., 2024a) are also limited to a single-room setting where most objects are visible to the agent.

Rearrangement has also been studied from the Task and Motion Planning (TAMP) perspective (Garrett et al., 2020a; 2021; 2020b). perspective. Garrett et al. (2020b) is limited to a single-room kitchen problem and assumes perfect detection of objects. Unlike most previous work, our proposed solution optimizes the traversal cost and addresses multi-room settings and imperfect object detection in an integrated POMDP framework. Tekin et al. (2023) and Mirakhor et al. (2024b) address the multi-room rearrangement problem. However, the decision-making process of Tekin et al. (2023) about when to explore and when to move an object is fixed and assumes perfect object detection. Our framework optimizes, naturally handles exploration and manipulation, and addresses object detector failures in a unified framework. Mirakhor et al. (2024b) make the assumption that objects are always on top of or inside containers. This limits its extendability to handling new problems, such as blocked paths where the objects could be in the path of another object without any containers. Our framework naturally allows for these new possibilities.

**Object Oriented POMDP (OOPOMDP) Planning:** Our work is partly inspired by Wandzel et al. (2019b), who OOPOMDP and perform a multi-object search in a 2D environment. Zheng et al. (2023) and Zheng et al. (2022) extend this formulation to object search in 3D environments. However, they are limited to the task of object search. In our work, we build on their formulation of OOPOMDP and extend it to include rearrangement actions and their corresponding belief updates. We further extend this rearrangement OO-POMDP to HOO-POMDP through action abstraction.

## 3 PROBLEM FORMULATION

**Environment and Agent**: Our agent is developed for the AI2Thor simulator environment (Kolve et al., 2017). It consists of a simulated house with a set of objects located in one or more rooms. The agent can take the following low-level actions: $A_s$ = ($MoveAhead$, $MoveBack$, $MoveRight$, $MoveLeft$, $RotateLeft$, $RotateRight$, $LookUp$, $LookDown$, $PickObject_i$, $PlaceObject$, $Start_{loc}$, $Done$). The $Move$ actions move the agent by a distance of $0.25m$ in the environment. The $Rotate$ actions rotate the agent pitch by 90 degrees. The $Look$ actions rotate the agent yaw by 30 degrees. $Start$ action starts the simulator and places the agent at the given location, and the $Done$ action ends the simulation. After executing any of the above actions, the simulator outputs the following information: a) RGB and Depth images, 2) the agent's position $(x, y, pitch, yaw)$, and 3) whether the action was successful. There are two types of objects in the world - *interactable* objects and *receptacle* objects. The *interactables* are the ones that can be picked and placed. The *receptacles* are objects that are not movable but can hold other objects.

**Task Setup:** Rearrangement is done in 2 phases. Walkthrough phase and rearrange phase. The walkthrough phase is meant to get information about stationary objects. The 2D occupancy map is generated in this phase, as well as the corresponding 3D Map. We get the size of the house(width and length) from the environment and uniformly sample points in the environment. We then take steps to reach these locations (if possible - some might be blocked). This simple algorithm ensures we explore the full house. At each of the steps, we receive the RGB and Depth. Using this, we create a 3D point cloud at each step and combine them all to get the overall 3D point cloud of the house with stationary objects. We then discretize this point cloud into 3D map voxels of size 0.25m, we further flatten this 3D map into a 2D map of grid cells (location in the 2D map is occupied if there exists a point at that 2D location at any height in the 3D map). While doing this traversal, we also get information about the receptacles by detector on the RGB images we receive during this traversal. This ends the walkthrough phase. (this walkthrough process needs to be done only once for any house configuration of stationary objects - walls, doors, tables, etc.). Then, objects are placed at random locations (done using AI2Thor environment reinitialization). This is when the rearrangement phase begins, with the planner taking the following as input - the map generated in the walkthrough phase, the set of object classes to move, and their goal locations.

# 4 HIERARCHICAL OBJECT ORIENTED POMDP (HOO-POMDP) PLANNER

This section presents our online planning framework designed to solve multi-object rearrangement problems in partially observable, multi-room simulated home environments. Our approach enables the system to tackle complex rearrangement tasks that involve challenging sub-problems, such as clearing blocked paths.

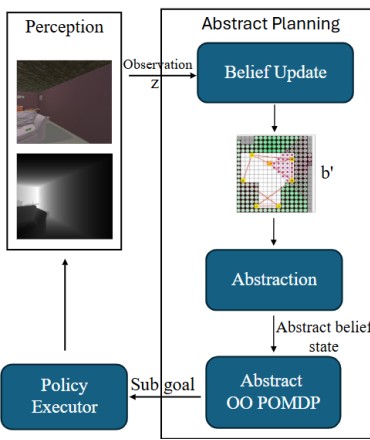

Figure 2: The agent receives RGB and depth images from environment at the start. The vision module creates the observation from this input and sends it to the belief update system. Belief is updated based on observation, and an abstract state is generated, which is sent to the OO POMDP Planner that outputs sub-goals. The sub-goals are used by the low-level policy executors to get and execute low-level actions in the environment.

**Overview:** Once the initial list of receptacles and 3D map have been generated, they, along with the goal information, are sent to the HOO-POMDP planner. The system operates in a cyclic fashion, integrating *perception*, *belief update*, *state abstraction*, *abstract planning*, and *action execution* (see Figure 2 and Algorithm 1). First, the *perception subsystem* detects objects in the RGB and depth image and outputs the observation $z$. This $z$ is used by the *belief update* subsystem to update the object-oriented belief state, which consists of the probability of each object being at a certain location in $M^{2D}$. The *abstraction system* uses this information to update its abstract state. The updated abstract state is sent to the abstract POMDP planner, which outputs a sub-goal that corresponds to a low-level policy. The low-level *policy executor* executes the low-level policy corresponding to the sub-goal. This might involve navigating to a specific location, grasping an object, or placing an object in a new position. After each action is executed, the environment state changes. The agent receives new output from the environment, and the cycle repeats until the overall rearrangement task is completed. In the rest of this section, we will discuss each of the subsystems and their interaction.

---

**Algorithm 1:** HOO-POMDP Rearrangement($M^{2D}, G, R$)

1 **Function** *HOO_POMDP_Rearrangement()*:
2    env ← InitializeEnv (); beliefState ← InitializeBeliefState ();
3    agent ← InitializeAgent (); loc ← random(); lowLevelAction ← Start$_{loc}$;
4    **while** *not TaskComplete()* **do**
5      *rgb, depth* ← env.GetObservation (*lowLevelAction*);
6      *observation* ← ProcessObservation (*rgb, depth*);
7      *beliefState* ← UpdateBelief (*beliefState, observation, lowLevelAction*);
8      *abstractState* ← GenerateAbstractState (*beliefState, G, R*);
9      *abstractAction* ← POUCTPlanner (*abstractState, beliefState*);
10      **if** *abstractAction == Done* **then**
11        return()
12      *lowLevelPolicy* ← SelectLowLevelPolicy (*abstractAction*);
13      *lowLevelAction* ← *lowLevelPolicy*.GetAction (*abstractAction, rgb, depth*);

---

**Abstract OO-POMDP Planning :**

**Background: POMDP:** A POMDP is a 7-tuple $(S, A, T, R, \gamma, O, O_{model})$ (Kaelbling et al., 1998). The state space $S$ is the set of states in which the agent and the objects in the environment can be. Action space $A$ is the set of actions that can be taken in the environment. The transition function

$T(s, a, s') = p(s'|s, a)$ is the probability of reaching the state $s'$ when the action $a$ is taken in the current state $s$. The agent receives an observation $z \in O$ when an action is taken. The probability of receiving an observation when being in a given state $s$ after having taken action $a$ is defined by the observation model $O_{model}(s, a, z) = p(z|s, a)$. The reward function $R(s, a)$ defines the reward received when taking action $a$ in state $s$, and $\gamma$ is the discount factor. In a partially observed world, the agent does not know its exact state and maintains a distribution over possible states, i.e., a belief state $b$. The belief is updated when an action is taken, and observation is received with the following equation, where $\eta$ is the normalizing constant:

$$b'(s') = \eta O(s', a, z) \sum_{s \in S} T(s, a, s')b(s) \tag{1}$$

**Object Oriented POMDP:** Object-oriented POMDP factors the state and observations over the objects. Each state $s$ is represented as a tuple of its $n$ objects $s = (s_1, \ldots, s_n)$, each observation $z = (z_1, \ldots, z_n)$ (Wandzel et al., 2019a) and the belief state $b$ is factorized as $b = \prod_{i=0}^{n} b_i$.

**Abstract Object Oriented POMDP:**

We now instantiate the rearrangement problem as an abstract POMDP. In our definition of the abstract OOPOMDP, we make an object independence assumption - that at any given time, the observation and belief of any object do not depend on any other object. More formally, $(P(z_i|s_j, z_j, s_i) = P(z_i|s_i)$, observation $z_i$ is independent of the states and observations of other objects, conditioned on its own state $s_i$ (observations are conditionally independent). Similarly, we also assume $P(s_i'|s_i, s_j, a) = P(s_i'|s_i, a)$ when $j! = i$, i.e., the next state of object $i$ only depends on its own previous state and the action. This allows us to represent the state and observation as entities factored on objects, which in turn helps make independent belief updates for each object(Algorithm 2).

- **State Space**: We use a factored state space that includes the robot state $s_r$, and the target object states $s_{targets}$. The complete state is represented as $s = (s_r, s_{targets})$. $s_{targets} = (s_{target_1}, \ldots, s_{target_n})$ where $n$ is the number of objects to be moved. $s_{target_i} = (loc_i, pick_i, place_{locs}, is\_held, at\_goal, g_i)$ : $loc_i$ is the current location of the object, $pick_i$ corresponds to the location from where this object can be picked, $place_{locs}$ corresponds to the set of locations (absolute 2D coordinates) from where this object can be placed from. $g_i$ is the goal location of the object. All locations are discretized grid coordinates in the 2D map ($M^{2D}$).

- **Action Space**: The action space consists of abstract navigation and interaction actions.
  $A = \{Move_{AB}, Rotate_{angle}, PickPlace_{\text{Object}_i - goal_{loc}}, Done\}$

- **Transition Model** :
    - $Move_{AB}$ - The move action moves the agent from location A to location B.
    - $Rotate_{angle}$ - The rotate action rotates the agent to a given angle.
    - $PickPlace_{\text{Object}_i - goal_{loc}}$ - The $PickPlace$ action picks $Object_i$ from the current position of the robot and places it at the given $goal_{loc}$.

- **Observation Space**: We use a factored observation space similar to state space factorization. Each observation can be divided into the robot observation and object observation $z = (z_{robot}, z_{objects})$, where $z_{objects} = (z_{target_1}, \ldots z_{target_n})$. Each observation $z_{target_i} \in L \cup \text{Null}$ - is a detection of the object $i$'s location or *Null* based on the detector's output for object $i$.

- **Observation model:** By definition of $z$ above, $\Pr(z|s) = \Pr(z_r|s_r)\Pr(z_{\text{objects}}|s_{targets})$ and $\Pr(z_r|s_r) = 1$ since the robot pose changes deterministically. Under the conditional independence assumption, $\Pr(z_{\text{objects}}|s)$ can be compactly factored as follows:

$$\Pr(z_{\text{objects}}|s) = \Pr(z_{target_1}, \ldots, z_{target_n}|s_{target_1}, \ldots s_{target_n}, s_r) \tag{2}$$

$$= \prod_{i=1}^{n} \Pr(z_{target_i}|s_{target_1}, \ldots, s_{target_n}, s_r), (\text{all } z_{target_i} \text{ are independent }) \tag{3}$$

$$= \prod_{i=1}^{n} \Pr(z_{target_i}|s_{target_i}, s_r), (z_{target_i} \text{ does not depend on \textbf{state} of other objects}) \tag{4}$$

$\Pr(z_i|s_{target_i}, s_r)$ is defined differently for each object based on the object detector's capability to detect the object of interest and the current state. More details are in A.1.2.

- **Reward Function**:
  - *$Move_{AB}$* : The cost of moving from location A to B [$Cost = -1 * N_a$ (where $N_a$ number of required actions)].
  - *$Rotate_{angle}$*: The cost of rotating the agent from the current rotation to the final given angle.
  - *$PickPlace_{Object_i - goal_{loc}}$* : Cost of moving from current location to goal location + cost of pick + cost of place. It gets an additional reward of 50 if the object is being placed at its goal location $g_i$ and this $g_i$ is free in the current state.
  - *Done* - This action receives a reward of 50 if all objects have been placed at their goal location and $-50$ otherwise.

**Abstract OOPOMDP Planner**: Given a task defined as an abstract OOPOMDP and an initial abstract state, the OOPOMDP planner uses partially observable UCT (POUCT) Silver & Veness (2010) to search through the space of abstract actions to find the best sub-goal. POUCT is an extension of the UCT algorithm Kocsis & Szepesvári (2006) to partially observable settings. The search tree in POUCT is over the histories instead of states. A history is a sequence of actions and observations $h_t = (a_1, z_1, \ldots, a_t, z_t)$. Similar to the UCT algorithm, the POUCT maintains a tree, where each node corresponds to a history $T(h)$, and for each node, a count variable $(N(h))$ and a value variable $(v(h))$ are stored, representing the number of times this history has been visited and the expected value of history $h$, which is estimated by the expected return of all simulations starting at $h$. The algorithm samples a state from the belief space $b$ for the current history, and if the tree already contains all the children for the current node, then the action with the best value is selected using the equation $V(ha) = V(ha) + c\sqrt{\frac{\log N(h)}{N(ha)}}$ to compute the value for all actions. If the tree does not contain all children, then a rollout policy is used to select actions for simulations, and the tree is updated with information from the simulations. Once that is done, the best action is selected and returned. The full algorithm is in Algorithm 3 in the Appendix. The rollout policy is a random policy that picks randomly amongst available actions. The *Move* and the *Rotate* actions are initialized by the values in the abstract state for each object. Recall that each object has the following information in its abstract state $s_i = (loc_i, pick_i, place_{locs}, is\_held, at\_goal)$. A separate $Move_{AB}$ is initialized with $A = agent\_pos$ and $B =$ all pick locations defined for all objects. $Rotate_{angle}$ - for all objects, less than 2m from the agent, the angle is computed based on the agent's required orientation to view the object from its current position. *PickPlace* - is defined for each object where the agent is less than 2m away from that particular object, for all locations in the $place_{locs}$ as $goal_{loc}$ initializing a set of *PickPlace* action for each object.

**Policy Executor:** Each sub-goal/action in the abstract OO-POMDP corresponds to a policy. When the planner outputs a sub-goal, the information in the given sub-goal is used to initialize the low-level policy. The output from the low-level policy is a sequence of low-level actions.

The *Move* sub-goal corresponds to the *Move* policy, which uses the $A^*$ algorithm to move from location A to B. The *Rotate* policy also uses the $A^*$ algorithm. The *PickPlace* policy consists of 2 RL agents and $A^*$ that picks the object from the current location and places it at the goal location.

- Sub-goal $Move_{AB}$ gives the *Move* policy the location B to move to from location $A$, which is used to initialize the $A^*$ algorithm and get a sequence of low-level move actions to reach goal location $B$. The action space available to the system is all the *Move* actions and all the rotate actions. It uses an Euclidian distance-based heuristic.

- Sub-goal $Rotate_{angle}$ gives the *Rotate* policy the final angle to be at, which is used as the final state the $A^*$ system must reach and outputs a sequence of rotate actions.

- Sub-goal $PickPlace_{Object_i - goal_{loc}}$ provides the object to interact with and which location to place it at. The policy takes this information as input and outputs a sequence of actions consisting of *Pick*, *Place*, and navigation actions that enable it to pick the object, move to the given destination, and place the object at that location. The *PickPlace* consists of 3 separate components a) An RL model trained to pick an object, b) the $A^*$ navigation model to go its destination c) An RL model trained to place the object when the agent is near the goal. All 3 of these run sequentially and make up the *PickPlace* Policy. It is designed this way to improve modularity and reduce

---

**Algorithm 2:** HOO-POMDP Belief Update

---

**1 Function** `UpdateBelief` (*beliefState, action, observation*) **:**

**2**    $b \leftarrow beliefState$ ; $z \leftarrow observation$;

**3**    **for** *each object $i$ in $b$* **do**

**4**      **for** *each possible state $s_{ij}$ of object $i$* **do**

**5**        **if** *action is navigation* **then**

**6**          $b'_i(s_{ij}) \leftarrow \eta p(z_i|s_{ij})b_i(s_{ij})$

**7**        **else if** *action is place* **then**

**8**          $b'_i(s_{ij}) \leftarrow 1$ if $s_{ij} =$ action.goalLocation, $0$ otherwise

**9**        **else if** *action is pick* **then**

**10**          $b'_i(s_{ij}) \leftarrow 1$ if $s_i =$ action.agentLocation, $0$ otherwise

**11**    **return** $b'$;

---

the complexity of each part. Both the pick model and place model are trained using the PPO algorithm. The action space is $\{A_s - Place\}$ for *pick* model and $\{A_s - Pick\}$ for the *place* model. The training process for the pick model involves randomly positioning the target object within a specified proximity to the agent. The goal is to pick a selected object successfully. For the place model, the training methodology follows a similar approach, with the key distinction being the absence of object detection requirements, as the agent begins each scenario already holding the object. In all training instances for the place model, the initial state consists of the agent holding an object, and the task involves depositing the object at a predetermined location.

**Perception:** Once the agent executes the low-level action, it receives an RGB and Depth image. An object detector is used to detect objects in the RGB image, and the depth map is used to get their $3D$ location in the world. This is used to generate the object-oriented observation $z = (z_1, \ldots, z_n)$.

**Belief Update**: Algorithm 2 presents the belief update function for our HOO-POMDP. The `UpdateBelief` function takes as input the current belief state $b$, the performed action $a$, and the received observation $z$. For each object $i$ in the belief state and each possible state $s_{ij}$ ($j = 1, \ldots, L$, where $L$ is the set of all its possible locations in the 2D Map) of that object, the algorithm updates the belief based on the action type. For navigation actions, it applies a probabilistic update using the observation model $p(z_i|s_{ij})$ (Line 6). For 'place' actions, it sets the belief to **1** if the object's state matches its desired goal location $g_i$, and **0** otherwise (Line 8). For 'pick' actions, it assigns a belief of **1** if the object's state corresponds to the agent's location and **0** otherwise (Line 10).

**Generating Abstract State:** We now have a belief state over the set of all possible locations for each object. We need to generate the abstract object-wise state consisting of object location information and their corresponding pick-and-place information. The information that needs to be computed for each object is as follows: $pick_i$, $place_{locs}$, $is\_held$, $at\_goal$.

The value for $is\_held$ comes from the previous low-level action and previous state. If the previous state had $is\_held$ as false and low-level action was to pick the object of interest, $is\_held$ is set to true. If the previous action was not a pick or a pick action for a different object, then the variable remains unchanged. If the previous action was $place$ and $is\_held$ is true, then it is set to False.

The value for $at\_goal$ is copied from the previous state if the last low-level action was not the place action. If it was, and if $is\_held$ was true in the previous state, then $at\_goal$ is set to true.

The value for $place_{locs}$ are sampled from the object goal location and three nearby receptacles as alternate goal locations for the object. For each of these goal locations, a location from where the object can be placed is sampled.

The location $pick_i$ is sampled based on the belief distribution of where the object could be and is the location from which the object can be picked. If the distribution over location is spread out, we sample multiple locations (by ensuring each sampled location is far from the other sampled locations for the same object). For both the locations in $place_{locs}$ and for the location $pick_i$, we then check if they are reachable. If they are not, those locations are discarded.

This sampling method enables our system to handle scenarios involving blocked goals, object swaps, and obstructed paths effectively. If an object's path is blocked, the planner will receive information indicating that there is no accessible location from which to pick up the object, necessitating the relocation of other objects first. When placing objects, we provide alternative receptacle locations. This approach allows us to move an object to another location if its goal position is blocked, thereby freeing up its current location. This strategy addresses both blocked goal and swap scenarios. Furthermore, this sampling process enhances our system's extensibility. We can incorporate additional constraints based on new object properties. For example, if opening an object requires interaction from a distance, the sampler can ensure that the sampled location is sufficiently far to enable successful opening.

After creating this abstract state, it is sent to the abstract planner, and the cycle starts again.

## 5 EXPERIMENTS

### 5.1 DATASET

- **RoomR:** This is the rearrangement challenge dataset proposed by Batra et al. (2020). It contains single-room environments with 5 objects to be rearranged. It has 25 room configurations with 40 different rearrangements for each room configuration.

- **ProcTHORRearrangement (Proc):** This is a dataset present in AI2Thor, which is bigger in terms of the rooms (two rooms, five objects) and, hence, partial observability. It has 125 room configurations with 80 rearrangements for each room configuration.

- **Multi RoomR:** We introduce a novel dataset designed to address more challenging problems, featuring larger environments (2-4 rooms) and an increased number of objects (10-20 objects). It has 400 room configurations. More details in Appendix A.2.4.

### 5.2 METRICS

- **Success Rate (SR): 1** if all objects have been moved to the correct goal locations, **0** otherwise.

- **Object Success Rate (OSR):** (Total Objects successfully moved)/(Total objects to move) - this metric captures the proportion of objects moved to the correct goal location.

- **Total Actions taken (TA):** A measure of the efficiency of the system in terms of the total number of actions taken. We present the average number(rounded up) of actions taken during successful runs where the scene was fully rearranged.

### 5.3 BASELINES DEFINITION

- **Perfect Knowledge (PK):** In this baseline, we will start with all the information about the world. That is, we know the initial locations of all the objects. This is the upper limit of the system's performance as there is no uncertainty to manage.

- **Perfect Detector with partial observability (PD):** In this baseline, we will be solving our multi-object rearrangement problem with a perfect detector (objects in the visual field are detected with 100% probability). The main challenge is to find all objects and move them around efficiently.

- **OURS (Imperfect Detector with partial observability):** In this setting, we will solve the multi-object rearrangement challenge where the agent is expected to handle perception uncertainty (the detector fails to detect objects in the visual field) along with object's starting position uncertainty.

- **OURS-HP** (ablation): In this setting, we remove the hierarchical planning and use the POMDP planner to output low-level actions directly.

**Experimental setup:** Each experimental setting was evaluated across 100 distinct rearrangement configurations. For the RoomR dataset, we utilized 25 different room setups, with four rearrangement configurations per setup. For the other datasets, we employed 100 unique room configurations, each with one rearrangement configuration. In the blocked goal and swap settings, at least one object's goal location was obstructed by another object. The swap case required the exchange of positions for at least one pair of objects to complete the rearrangement task. In the blocked path scenario, all scenes contained a minimum of one object that needed to be moved to its goal position from its initial location to enable the rearrangement of other objects.

| Dataset | Objs | #BG | #Sw | #BP | #Rm | #V | PK | | | PD | | | Ours | | | OURS-HP | | |
|---|---|---|---|---|---|---|---|---|---|---|---|---|---|---|---|---|---|---|
| | | | ap | | | | SS↑ | OSR↑ | TA↓ | SS↑ | OSR↑ | TA↓ | SS↑ | OSR↑ | TA↓ | SS↑ | OSR↑ | TA↓ |
| RoomR | 5 | 1 | 0 | 0 | 1 | 3-4 | **63** | **88** | **176** | 62 | 87 | 189 | 49 | 71 | 211 | 13 | 33 | 302 |
| | | 2 | 1 | 1 | 1 | 3-4 | **51** | **77** | **210** | 49 | 76 | 238 | 39 | 61 | 289 | 8 | 27 | 392 |
| Proc | 5 | 1 | 0 | 0 | 2 | 2-3 | **60** | **82** | **203** | 60 | 81 | 269 | 46 | 68 | 352 | 9 | 29 | 410 |
| | | 2 | 1 | 1 | 2 | 2-3 | **47** | **71** | **246** | 42 | 63 | 311 | 31 | 53 | 398 | 4 | 19 | 565 |
| Multi RoomR | 10 | 1 | 1 | 0 | 2 | 2-3 | **41** | **77** | **457** | 40 | 79 | 529 | 32 | 65 | 710 | 5 | 25 | 1029 |
| | | 2 | 1 | 1 | 2 | 2-3 | **33** | **69** | **489** | 29 | 67 | 587 | 21 | 49 | 789 | 2 | 19 | 1092 |
| | 10 | 2 | 1 | 0 | 3-4 | 1-2 | **39** | **75** | **726** | 37 | 74 | 834 | 30 | 62 | 1189 | 3 | 16 | 1408 |
| | | 2 | 1 | 1 | 3-4 | 1-2 | **32** | **69** | **789** | 31 | 72 | 985 | 18 | 44 | 1321 | 1 | 7 | 1549 |
| | 15 | 1 | 1 | 0 | 3-4 | 2-3 | **32** | **78** | **895** | 30 | 74 | 921 | 22 | 59 | 1228 | * | * | * |
| | | 2 | 1 | 1 | 3-4 | 2-3 | **29** | **71** | **988** | 25 | 69 | 965 | 14 | 41 | 1416 | * | * | * |
| | 20 | 2 | 1 | 0 | 3-4 | 2-4 | **27** | **75** | **1168** | 27 | 74 | 1197 | 17 | 55 | 1621 | * | * | * |
| | | 2 | 1 | 1 | 3-4 | 2-4 | **22** | **70** | **1307** | 20 | 68 | 1336 | 10 | 36 | 1786 | * | * | * |

Table 1: Comparison between our methods with different levels of information and difficulty of problems and ablation on hierarchy. The difficulty is represented in terms of the following: a) **#BG** - number of blocked goal locations, b) **#Swap** Number of objects that need to be swapped, c) **#BP**: Number of objects blocking the path that need to be moved out of the way, d) **#Rm**: Number of rooms in the environment, e) **#V**: Number of objects initially visible. **\*** indicates results not ready yet. Will be in by camera-ready version.

## 6 RESULTS AND DISCUSSION

**Methods comparison:** The perfect knowledge system is the best performance our agent can have - there is no uncertainty in the world, and hence, it boils down to a path-traversal length optimization problem. We can see from Table 1 that the results of the perfect detector setting and the perfect knowledge setting are similar in the overall scene success rate as well as the object success rate. This shows that the PPOMDP planner is able to explore and find all the objects perfectly. The failures in both cases are due to the low-level policies failing to pick/place objects. The difference between the methods can be seen in the steps taken. The perfect detector system has to explore and rearrange and hence ends up taking more steps than the perfect knowledge system. The third set of results are for our system with an imperfect detector and partial observability. We can see that the detector failure causes our success rate to fall by a fair amount - this is because if an object is not detected more than once at its location, the planner's belief update prevents it from going to that region to pick that object up. Since our detector only has a 50-60% success rate (different for different classes of objects), it misses a fair subset of the objects, and hence the drop. As a result of needing to explore more due to detector failure, the number of steps taken is greater than that of the other methods. The difference between OURS and OURS-HP clearly shows the importance of hierarchy and abstraction. The performance drop is significant across all settings for OURS-HP.

**Comparison Across Datasets:** We can see that the results of the RoomR dataset of a single room and the Proc dataset with multiple rooms are similar (in scene success rate and object success rate). This indicates that our method effectively manages the exploration of multiple rooms as well as a reduction in the number of objects visible initially. Initial object visibility decreases when scaling from 2 to 4 rooms in MultiRoomR with ten objects, yet success rates remain relatively stable. The performance drops in our custom dataset with a larger number of objects. In particular, the object success rate drops slightly, but the scene success rate drops significantly. This is because, for a scene to succeed, we need all objects to be rearranged correctly, so even if 9/10 objects are rearranged correctly, it is still considered a failure. Hence, there is a bigger reduction in scene success rate.

**Across Different Challenges:** From the different rows for each dataset (the difference being the existence of blocked paths), we can see that the results for the blocked path version of the problem are lower. This is because the low-level policy is not as good at picking the objects on the floor as it is at picking the objects from the top of other objects. Also, the number of PickPlace actions increases as we tackle more complex problems such as swaps.

**Error Analysis:** The majority of failures of our system are due to low-level policy failures - the pick or the place action fails due to the imperfections of the low-level RL policy. The other type of failure in our system is due to belief estimation errors caused by detector failures - if a detector is looking at a certain location and it fails to detect an object multiple times (due to partial occlusion

or high distance), then our belief about the object being in that region reduces to a large extent. In that case, we are unlikely to come back to that part of the house. If a false positive happens, an agent might pick up the wrong object and place it and believe it has placed the correct object - leading to object and scene failure. These are the main causes of our failures in imperfect detector settings.

**Comparison to existing baselines:** It is important to note that our system addresses a variant of the multi-object rearrangement problem that differs in key aspects from those tackled by existing baselines. Unfortunately, this means that direct comparisons are not meaningful or informative. The primary distinction lies in the prior knowledge available to our system: we are given information about the classes of objects to be moved, whereas other systems Mirakhor et al. (2024a) (Mirakhor et al., 2024b), (Gadre et al., 2022) operate without this advantage. The motivation for it is that when this information is not provided, agents must perform a walkthrough phase for each new goal configuration to identify movable objects. In contrast, our formulation requires only a single initial walkthrough to map stationary objects in the environment. Subsequently, our system can efficiently handle multiple goal configurations without additional walkthroughs.

However, it is worth highlighting that our problem formulation introduces its own set of challenges. In particular, while existing systems report initial visibility of approximately 60% ((Mirakhor et al., 2024b), table 1) of target objects at the outset of their tasks, only about 20% of the objects are initially visible in our problem settings, necessitating more extensive and strategic exploration. This reduced initial visibility significantly increases the complexity of our task in terms of efficient exploration and belief management. It underscores the importance and effectiveness of our hierarchical planning approach in handling partial observability and perception uncertainty. The other key difference is we use low-level policies for navigation and manipulation, whereas other works Mirakhor et al. (2024b) and Mirakhor et al. (2024a) assume perfect navigation and manipulation capabilities.

While the difficulties faced by our system and existing baselines are not directly comparable, we believe our results demonstrate the efficacy of our approach in solving a challenging and practically relevant variant of the multi-object rearrangement problem. The performance across various metrics and scenarios, as detailed in the preceding sections, showcases the robustness and scalability of our HOO-POMDP framework in environments with high uncertainty and limited initial information.

**Limitations:** Our system makes an object independence assumption - that the state and observation of one object are not dependent on the state of other objects. However, this assumption doesn't always hold in cluttered environments. In such cases, observing, picking up, or placing an object might be affected by the positions of nearby objects. As a result, our object-oriented belief update system would need to be modified to handle these object interactions in more complex scenarios. HOO-POMDP cannot handle an unknown class of objects. It could potentially be handled by categorizing all of the known object types into a single 'unknown' class. The difficult part, however, is to plan to find an empty space to move the unknown object to. In the worst case, this could lead to complicated packing problems that are NP-hard.

## 7 CONCLUSION AND FUTURE WORK

In this paper, we presented a novel Hierarchical Object-Oriented POMDP Planner (HOO-POMDP) for solving multi-object rearrangement problems in partially observable, multi-room environments. Our approach decomposes the complex task into a high-level abstract POMDP planner for generating sub-goals and low-level policies for execution. Key components include an object-oriented state representation, belief updating to handle perception uncertainty, and an abstraction system to bridge the gap between continuous and discrete planning. Experimental results across multiple datasets demonstrate the effectiveness of our approach in handling challenging scenarios such as blocked paths and goals. The HOO-POMDP framework showed robust performance in terms of success rate and efficiency comparable to Oracle baselines with perfect knowledge or perfect detection. Notably, our method scaled well to environments with more objects and rooms. One of the ways to expand the scope is to relax the assumption of object independence partially. We can allow objects to be dependent on a small number of objects (e.g., objects in their close vicinity). Belief updates can now consider a small set of objects at any time. This relaxation helps maintain the efficient belief update while accounting for more real-world situations such as object-object interaction. Another potential future work is to handle stacking of objects and more cramped spaces, where more careful reasoning about object interactions is needed to plan the actions and order them appropriately.

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

## A  APPENDIX

### A.1  OBJECT DETECTION

#### A.1.1  DETECTION MODEL : YOLOV10

We collect data from the AI2Thor simulator. We do this by placing the agent in random locations in 500 scenes and extracting the RGB images along with the ground truth object bounding box annotations from the simulator. We have $50$ pickupable object classes in all our scenes combined. We train the YoloV10 detector on 10,000 images collected from these 500 scenes.

#### A.1.2  OBSERVATION PROBABILITY FOR POMDP

The probability of each individual observation based on the current state is the following.

$$\Pr(z_i|s_{target_i}, s_r) = \begin{cases} 1.0 - \text{TP} & s_i \in \mathcal{V}(s_{\text{robot}}) \wedge z_i = \text{null} \\ \delta\text{FP}/|\mathcal{V}_E(r)| & s_i \in \mathcal{V}(s_{\text{robot}}) \wedge \|z_i - s_i\| > 3\sigma \\ \delta z_i & s_i \in \mathcal{V}(s_{\text{robot}}) \wedge \|z_i - s_i\| \leq 3\sigma \\ 1.0 - \text{FP} & s_i \notin \mathcal{V}(s_{\text{robot}}) \wedge z_i = \text{null} \\ \delta\text{FP}/|\mathcal{V}_E(r)| & s_i \notin \mathcal{V}(s_{\text{robot}}) \wedge z_i \neq \text{null} \end{cases}$$

The detection model is parameterized by

- TP: Is the True positive of the Detection model for object class $i$.
- FP: Is the False positive of the Detection model for object class $i$.
- $r$: is the average distance between the agent and the object for true positive detections.

Table 2: Performance Metrics by Class : TP = True Positive, FP = False positive, r = average distance

| Class | r (m) | TP | FP |
|---|---|---|---|
| AlarmClock | 3.010 | 0.383 | 0.022 |
| Apple | 3.298 | 0.065 | 0.002 |
| BaseballBat | 2.941 | 0.499 | 0.011 |
| BasketBall | 2.631 | 0.336 | 0.003 |
| Book | 2.888 | 0.535 | 0.101 |
| Bottle | 2.733 | 0.465 | 0.006 |
| Bowl | 2.695 | 0.448 | 0.073 |
| Box | 3.977 | 0.225 | 0.012 |
| Bread | 1.523 | 0.082 | 0.010 |
| ButterKnife | 2.084 | 0.156 | 0.009 |
| CD | 2.082 | 0.085 | 0.001 |
| Candle | 3.325 | 0.048 | 0.004 |
| CellPhone | 2.137 | 0.327 | 0.006 |
| CreditCard | 1.107 | 0.042 | 0.002 |
| Cup | 2.505 | 0.513 | 0.012 |
| DishSponge | 1.714 | 0.306 | 0.003 |
| Kettle | 2.655 | 0.415 | 0.001 |
| KeyChain | 1.725 | 0.154 | 0.007 |
| Knife | 1.226 | 0.056 | 0.002 |
| Ladle | 2.333 | 0.015 | 0.000 |
| Laptop | 3.405 | 0.605 | 0.019 |
| Lettuce | 2.681 | 0.336 | 0.003 |
| Mug | 2.734 | 0.529 | 0.010 |
| Newspaper | 2.286 | 0.264 | 0.005 |
| Pan | 2.757 | 0.350 | 0.012 |
| PaperTowelRoll | 3.066 | 0.338 | 0.018 |
| Pen | 2.471 | 0.081 | 0.013 |
| Pencil | 1.853 | 0.040 | 0.015 |
| PepperShaker | 2.042 | 0.310 | 0.016 |
| Pillow | 3.615 | 0.683 | 0.037 |
| Plate | 2.278 | 0.355 | 0.010 |
| Plunger | 2.900 | 0.745 | 0.005 |
| Pot | 4.064 | 0.417 | 0.010 |
| Potato | 2.138 | 0.157 | 0.004 |
| RemoteControl | 2.176 | 0.324 | 0.025 |
| SaltShaker | 1.940 | 0.098 | 0.010 |
| SoapBottle | 3.147 | 0.519 | 0.038 |
| Spatula | 1.443 | 0.134 | 0.002 |
| SprayBottle | 2.744 | 0.085 | 0.031 |
| Statue | 3.095 | 0.650 | 0.033 |
| TeddyBear | 3.093 | 0.417 | 0.003 |
| TennisRacket | 3.111 | 0.128 | 0.017 |
| TissueBox | 4.087 | 0.286 | 0.003 |
| ToiletPaper | 2.806 | 0.383 | 0.006 |
| Vase | 3.230 | 0.699 | 0.095 |
| Watch | 1.661 | 0.210 | 0.006 |
| WineBottle | 2.903 | 0.667 | 0.003 |

- $V_E(r)$: It is the visual field of view of 90 degrees within distance $r$.

- $\delta$ : is the distance weight, it is 1 if detection is within $V_E(r)$, else $\delta = 1/d$, where $d$ is the distance from the robot to the object.

The list of $TP$, $FP$ and $r$ for the object classes in the dataset presented in table 2.

## A.2 MULTIROOMR DATASET DETAILS

### A.2.1 OVERVIEW

The dataset consists of 400 distinct room configurations, with varying complexity in terms of room count and object arrangements.

### A.2.2 DATASET COMPOSITION

- **Total Size:** 400 room configurations.
- **Object Types:** Comprehensive selection from AI2Thor environment (see Appendix).
- **Object Selection Criteria:** Includes the majority of AI2Thor objects, excluding objects too small for reliable detection even at close range.

### A.2.3 ROOM CONFIGURATION DISTRIBUTION

Two-Room Configurations

- Total configurations: 200
- Objects per configuration: 10
- Path characteristics:
  - 50% contain blocked paths.
  - 10 rearrangements per configuration.

Three-Room Configurations

- Total configurations: 100
- Path characteristics:
  - 50% contain blocked paths.
  - 30 rearrangements per configuration.
  - Distribution: 10 rearrangements each for 10, 15, and 20 objects.

Four-Room Configurations

- Total configurations: 100
- Path characteristics:
  - 50% contain blocked paths.
  - 30 rearrangements per configuration.
  - Distribution: 10 rearrangements each for 10, 15, and 20 objects.

### A.2.4 OBJECT PLACEMENT CRITERIA

1. **Room-wide Movement Requirement:** Each room must contain at least one object requiring movement, ensuring comprehensive exploration by the agent.
2. **Blocking and Swapping Scenarios:** Configurations include:
   - Objects blocking goal locations of other objects.
   - Objects mutually blocking each other's goals (swap cases).
3. **Path Blocking Optimization:** In scenes with blocked paths, blocking objects are strategically placed to maximize inaccessible house area such that at least one object must be moved out of the way to access all objects.

## A.3 POUCT

---

**Algorithm 3:** POUCT Planner (Modified version from Silver & Veness (2010))

---

**1 Function** *POUCTPlanner(abstractState, beliefState)***:**

  **2**    $b \leftarrow beliefState$;

  **3**    $abs \leftarrow abstractState$;

  **4**    $T \leftarrow \{\}$;

  **5**    **for** $j = 0$ **to** $SIMULATIONS$ **do**

  **6**       $\hat{s} \leftarrow$ SAMPLE $(b)$;

  **7**       SIMULATE $(\hat{s}, \{\}, 0, abs)$;

  **8**    $abstractAction \leftarrow \arg\max_a V(ha)$;

  **9**    **return** $abstractAction$;

**10 Function** SAMPLE $(b)$**:**

  **11**    **for** $o \in Obj$ **do**

  **12**       $\hat{s}_o \sim b_o$;

  **13**    **return** $\bigcup \hat{s}_o$;

**14 Function** SIMULATE $(s, h, depth, abs)$**:**

  **15**    **if** $\gamma^{depth} < \epsilon$ **then**

  **16**       **return** 0;

  **17**    **if** $h \notin T$ **then**

  **18**       **for** *all* $a \in \mathcal{A}$ **do**

  **19**          $T(ha) \leftarrow \langle N_{init}(ha), V_{init}(ha) \rangle$;

  **20**       **return** ROLLOUT$(s, h, depth, abs)$;

  **21**    $a \leftarrow$ selectMaxAction();

  **22**    $(s', z, r) \sim \mathcal{G}(s, a)$;

  **23**    $R \leftarrow r + \gamma \cdot$ SIMULATE $(s', hao, depth + 1)$;

  **24**    $T(ha) \leftarrow \left\langle N(ha) + 1, V(ha) + \frac{R - V(ha)}{N(ha)} \right\rangle$;

  **25**    **return** $R$;

**26 Function** ROLLOUT $(s, h, depth, abs)$**:**

  **27**    **if** $\gamma^{depth} < \epsilon$ **then**

  **28**       **return** 0;

  **29**    $a \sim \pi_{rollout}(h, abs)$;

  **30**    $(s', o, r) \sim \mathcal{G}(s, a)$;

  **31**    **return** $r + \gamma \cdot$ ROLLOUT $(s', hao, depth + 1)$;

---

## A.4 QUALITATIVE RESULTS

We test our system with different depths for the MCTS planner to see how much look-ahead affects the performance of our system. MCTS search depth is one of the important factors determining the amount of exploration and the time taken for search at step. Results are shown in table 3. The depth for the system - OURS is 12, and the depth for OURS-MCTS_1 is 1. From the table, we can see that the greedy approach of just looking ahead by 1 step is not enough to solve the rearrangement problem. This is because, when we look ahead only 1 step, we do not get enough reward/feedback from the world about what is a good path to take and hence makes it extremely hard to solve the problem. We can also see that, in the problems it does solve, it takes significantly longer to solve. We did not compute the results for 15 and 20 objects in this setting as it failed to solve any problems in 3-4 rooms with ten objects.

| Dataset | Objs | #BG | #Sw | #BP | #Rm | #V | | Ours | | | | OURS-MCSTS_1 | | |
| | | ap | | | | | | SS↑ | OSR↑ | TA↓ | Time(m)↓ | SS↑ | OSR↑ | TA↓ |
|---------|------|-----|-----|-----|-----|-----|---|------|------|------|----------|------|------|------|
| RoomR | 5 | 1 | 0 | 0 | 1 | 3-4 | | **49** | **71** | **211** | 1.61 | 8 | 26 | 565 |
| | | 2 | 1 | 1 | 1 | 3-4 | | **39** | **61** | **289** | 2.15 | 6 | 21 | 721 |
| Proc | 5 | 1 | 0 | 0 | 2 | 2-3 | | **46** | **68** | **352** | 3.42 | 2 | 12 | 875 |
| | | 2 | 1 | 1 | 2 | 2-3 | | **31** | **53** | **398** | 4.52 | 1 | 10 | 1102 |
| Multi RoomR | 10 | 1 | 1 | 0 | 2 | 2-3 | | **32** | **65** | **710** | 7.89 | 0 | 7 | 0 |
| | | 2 | 1 | 1 | 2 | 2-3 | | **21** | **49** | **789** | 8.98 | 0 | 3 | 0 |
| | 10 | 2 | 1 | 0 | 3-4 | 1-2 | | **30** | **62** | **1189** | 11.45 | 0 | 0 | 0 |
| | | 2 | 1 | 1 | 3-4 | 1-2 | | **18** | **44** | **1321** | 12.97 | 0 | 0 | 0 |
| | 15 | 1 | 1 | 0 | 3-4 | 2-3 | | **22** | **59** | **1228** | 16.89 | NC | NC | NC |
| | | 2 | 1 | 1 | 3-4 | 2-3 | | **14** | **41** | **1416** | 17.12 | NC | NC | NC |
| | 20 | 2 | 1 | 0 | 3-4 | 2-4 | | **17** | **55** | **1621** | 21.61 | NC | NC | NC |
| | | 2 | 1 | 1 | 3-4 | 2-4 | | **10** | **36** | **1786** | 23.79 | NC | NC | NC |

Table 3: Performance metrics for our method across different depths for MCTS search. Metrics: Success Score (SS), Object Success Rate (OSR), Task Actions (TA), execution Time in minutes, and number of objects initially visible (#V). The difficulty parameters include the number of blocked goals (#BG), objects to be swapped (#Sw), blocking objects (#BP), and number of rooms (#Rm). NC: Not computed.

