# OpenReview forum: "Hierarchical Object-Oriented POMDP Planning for Object Rearrangement"
_ICLR.cc/2025/Conference — Submitted to ICLR 2025_

### Official Review · Reviewer_Jcan · 2024-11-04

**Soundness:** 2
**Presentation:** 2
**Contribution:** 2
**Rating:** 5
**Confidence:** 5

**Summary:**

The paper deals with the problem of Multi-room rearrangement using a Hierarchical POMDP approach. The problem is difficult as it involves a number of difficulties including combinatorial expansion in complexity with increasing number of objects, partial observability due to limited field of view, scalability etc. The paper tackles the partial observability by maintaining an object oriented belief state to account for the possible locations of the objects. From this belief, the object state is abstracted which indicates whether the object is picked, placed, is held etc. Based on this state space for all the objects, the POMDP planner generates a high-level action to be executed such as the PickPlace, Move, Rotate etc. These high-level actions are then executed by the low-level policies which are heuristic as well as RL based. The paper claims to achieve a scalable and efficient rearrangement plan in Multi-room rearrangement scenarios. The paper also introduces a novel dataset MultiRoom R.

**Strengths:**

- The paper tackles a very complex and understudied problem of Multi-room rearrangement.
- Usage of POMDP based planners to effectively address uncertainities in a large multi-room space is very interesting.
- The supplementary video was good.
- The paper also presents a new dataset Multi RoomR which includes "blocked path scenarios" as an additional complexity in rearrangement.

**Weaknesses:**

- **Uncertainty over objects start locations :** As stated in the paper line 338-339 : “The location $pick_i$ is sampled based on the belief distribution of where the object could be…” So for an unseen object, is the initial belief distribution a uniform probability over the entire 2D map? If yes, doesn’t it imply that your planner starts execution based on a randomly sampled location? Doesn’t it make planning less effective? Moreover, for predicting $loc_i$ of each object, wouldn’t the use of any common sense priors make the $loc_i$ prediction better and lead to faster convergence?
    - Don’t you think it would be better to use some commonsense knowledge about the object-receptacle-room relationships as a bias for belief initialization and update? Because in indoor scenarios, multiple methods have shown that the application of commonsense knowledge for indoor environments aids in planning - Sarch et al.[1], Kant et al.[2] and Mirakhor et al. [3,4].
-  **Comparison Results :** This paper shows no comparison study with the existing state-of-the art (SOTA) methods. As stated in Sec 2 : Related Work - Mirakhor et al. [4] addresses the multi-room rearrangement problem. Even other rearrangement methods that show results in single room rearrangement such as Gadre et al.[5], Sarch et al.[1] and Trabucco et al.[6] should be compared with to ground the claims of this paper. In fact, all these methods have shown their results in Ai2Thor. As stated in Line 507-508 the author mentions that they have an advantage over the SOTA method because “classes of objects to be moved are known” - Is it known for every rearrangement scenario, if so why do you need it? Or Is this method limited to only these classes of objects? Please clarify. Also, as you stated in Line 509-513 that your problem initialization is much more difficult due to initial object visibility. What is the reference or study for the claim on the existing methods initial visibility being approximately 60%? Please show empirically how difficult the problem becomes with variations in the percentage of initial object visibility.
- **Scalability :** To understand whether this method is scalable to an increasing number of objects and rooms, more results need to be shown with the number of objects varying from 5, 10, 15 say up to 20 on the same dataset. Similar results can be shown with the number of rooms varying from 2,3,4 & 5, keeping the number of objects constant. Presently, the paper shows results for only 5 objects on RoomR and Procthor, whereas it shows results for 10 objects on Multi-RoomR. This makes it difficult to establish a trend for results with an increasing number of objects and rooms.
- **Ablation study :** The two baselines - PK and PD used in the paper study only the perception efficacy, what about the planning efficacy? Can you replace your planner with some alternatives such as a classical traveling salesman problem (TSP) solver, an optimizer based OR-Tools[7] planner, a greedy planner etc. This will give an insight of how close to the optimal is this planner and how much of an improvement is this method over the heuristic strategies.
- **Novel Dataset (Multi RoomR) :** As far as I know, ProcThor has multi-room scenarios with up to 5 rooms and about 20 objects. But, the authors have stated in Line 418-420 about ProcThor having - “2 rooms, 5 objects”. Are you sure about this? This begs the question regarding the motivation of the new dataset - Multi RoomR? What was missing in ProcThor? What is the object, receptacle, room type distribution in the new dataset? How do we gauge the complexity of this new dataset, if there are no comparison results with SOTA methods?
- **Metrics and problem configurations :** To show the efficacy of planning, will it now be more beneficial to show the time and distance the agent took to solve the entire task? Moreover, to highlight the difficulty of partial observability, can you specify how many or what percent of objects are initially visible and how many actions or time does the agent take to find them?

[1] Gabriel Sarch, Zhaoyuan Fang, Adam W. Harley, Paul Schydlo, Michael J. Tarr, Saurabh Gupta, and Katerina Fragkiadaki. 2022. TIDEE: Tidying Up Novel Rooms Using Visuo-Semantic Commonsense Priors. In Computer Vision – ECCV 2022.\
[2] Kant, Y.; Ramachandran, A.; Yenamandra, S.; Gilitschenski, I.; Batra, D.; Szot, A.; and Agrawal, H., Housekeep: Tidying Virtual Households using Commonsense Reasoning. In European Conference on Computer Vision, 2022.\
[3] Karan Mirakhor, Sourav Ghosh, Dipanjan Das, and Brojeshwar Bhowmick. Task Planning for Visual Room Rearrangement under Partial Observability. In The Twelfth International Conference on Learning Representations, 2024.\
[4] Karan Mirakhor, Sourav Ghosh, Dipanjan Das, and Brojeshwar Bhowmick.Task Planning for Object Rearrangement in Multi-Room Environments. In Proceedings of the AAAI Conference on Artificial Intelligence, volume 38, pp.10350–10357, 2024.\
[5] Gadre, S.Y., Ehsani, K., Song, S., & Mottaghi, R. (2022). Continuous Scene Representations for Embodied AI. 2022 IEEE/CVF Conference on Computer Vision and Pattern Recognition (CVPR), 14829-14839.\
[6] Brandon Trabucco and Gunnar A Sigurdsson and Robinson Piramuthu and Gaurav S. Sukhatme and Ruslan Salakhutdinov, A Simple Approach for Visual Room Rearrangement: 3D Mapping and Semantic Search, The Eleventh International Conference on Learning Representations, 2023.\
[7] Laurent Perron and Frédéric Didier, Google, https://developers.google.com/optimization/cp/cp_solver/

**Questions:**

Same as the Weaknesses

---

> ### Author Response · Authors · 2024-11-17
> **Response part 1**
>
> We thank the reviewer for these constructive comments.
>
> >W1: “Don’t you think it would be better to use some commonsense knowledge about the object-receptacle-room relationships as a bias for belief initialization and update? Because in indoor scenarios, multiple methods have shown that the application of commonsense knowledge for indoor environments aids in planning - Sarch et al.[1], Kant et al.[2] and Mirakhor et al. [3,4].”
>
> **Common-sense priors:**
> Yes, for an unseen object, the planner starts execution based on a randomly sampled location.  While common-sense priors can help, a common application of rearrangement is cleaning homes, and untidy homes may only sometimes follow common-sense priors. Our system can handle all cases where objects can be anywhere with similar efficiency.  It is also **easy to add common-sense priors** to our framework (by changing initial belief from random to initialized based on common-sense priors) if we have more information about the domain we operate in (for ex: warehouse).
>
> >W2: “​​ This paper shows no comparison study with the existing state-of-the art (SOTA) method.  …. What is the reference or study for the claim on the existing methods initial visibility being approximately 60%? Please show empirically how difficult the problem becomes with variations in the percentage of initial object visibility.”
>
> >W6: “Metrics and problem configurations : To show the efficacy of planning, will it now be more beneficial to show the time and distance the agent took to solve the entire task? Moreover, to highlight the difficulty of partial observability, can you specify how many or what percent of objects are initially visible and how many actions or time does the agent take to find them?”
>
> **Comparison to SOTA methods**
> It is important to note that our system addresses a **variant of the multi-object rearrangement problem** that differs in key aspects from those tackled by prior works (Gadre et al.[5], Sarch et al.[1], Trabucco et al.[6], and Mirakhor et al.[4]). While our approach leverages prior knowledge of target object classes, this design choice enables broader generalization capabilities. In existing approaches, **where such information is not provided, agents must perform a walkthrough phase for each new goal configuration to identify movable objects**. In contrast, our formulation **requires only a single initial walkthrough** to map stationary objects in the environment. Subsequently, our system can **efficiently handle multiple goal configurations** without additional walkthroughs, significantly enhancing its adaptability to diverse scenarios. This fundamental difference in problem formulation makes direct performance comparisons potentially misleading despite operating in similar environments.
>
> We acknowledge this limitation in direct comparability but believe our results demonstrate the effectiveness of our approach in solving a practically relevant variant of the rearrangement problem. We will revise Section 6 to make these distinctions clearer and better contextualize our contributions relative to prior work.
>
> **Clarification on Object Classes**: - We do not restrict which objects can be moved - Rather, for each specific rearrangement task, the system needs to know which object classes are targets - This allows flexibility while maintaining efficient exploration and planning.
>
> For the claim about 60%, **it refers to  MiraKhor et al., table 1**, where  #V / #O gives the initially visible percentage, which is 60% for all settings in their settings. We will add this information in the paper along with the visibility percentage of objects in our datasets.
>
> **New Metrics**: We will also add a separate table in the appendix with results on how much time was taken to complete the tasks.The distance the agent took to solve the problem is directly proportional to the total actions taken and, hence, not very beneficial in providing any new insight into the method's effectiveness.
>
> >W3: “To understand whether this method is scalable to an increasing number of objects and rooms, more results need to be shown with the number of objects varying from 5, 10, 15 say up to 20 on the same dataset…. This makes it difficult to establish a trend for results with an increasing number of objects and rooms.”
>
> **Scalability**: We currently have results for 2,3,4 rooms with the same number of objects (10) in the MultiRoomR dataset.  RoomR and ProcThor dataset results also show this - both datasets have 5 objects but are different in the number of rooms (1-2).  This shows how the method behaves with different numbers of rooms with the same number of objects.
>
> We will provide results for larger numbers of objects and add them in the paper (up to 20)

---

> > ### Comment · Reviewer_Jcan · 2024-11-27
> >
> > I thank the authors for their responses and appreciate their effort.
> >
> >     Common-sense priors: Yes, for an unseen object, the planner starts execution based on a randomly sampled location. While common-sense priors can help, a common application of rearrangement is cleaning homes, and untidy homes may only sometimes follow common-sense priors. Our system can handle all cases where objects can be anywhere with similar efficiency. It is also easy to add common-sense priors to our framework (by changing initial belief from random to initialized based on common-sense priors) if we have more information about the domain we operate in (for ex: warehouse).
> >
> > If it is easy to integrate the belief model into the pipeline, I request the authors to kindly show some results for the method with commonsense prior based initial belief v/s random initial belief.
> >
> >     Comparison to SOTA methods It is important to note that our system addresses a variant of the multi-object rearrangement problem that differs in key aspects from those tackled by prior works (Gadre et al.[5], Sarch et al.[1], Trabucco et al.[6], and Mirakhor et al.[4]). While our approach leverages prior knowledge of target object classes, this design choice enables broader generalization capabilities. In existing approaches, where such information is not provided, agents must perform a walkthrough phase for each new goal configuration to identify movable objects. In contrast, our formulation requires only a single initial walkthrough to map stationary objects in the environment. Subsequently, our system can efficiently handle multiple goal configurations without additional walkthroughs, significantly enhancing its adaptability to diverse scenarios. This fundamental difference in problem formulation makes direct performance comparisons potentially misleading despite operating in similar environments.
> >
> >     We acknowledge this limitation in direct comparability but believe our results demonstrate the effectiveness of our approach in solving a practically relevant variant of the rearrangement problem. We will revise Section 6 to make these distinctions clearer and better contextualize our contributions relative to prior work
> >
> > I understand that the method proposed can perform multiple goal configuration. However, from my understanding it is clear that the method can perform single goal configuration as well and that is what the existing methods do. Making the experimental setup for single goal configuration, the comparison study will tell us how the proposed method compares to the existing methods in a traditional rearrangement setting with goal stage walkthrough and shuffle phase rearrangement.
> >
> >     Clarification on Object Classes: - We do not restrict which objects can be moved - Rather, for each specific rearrangement task, the system needs to know which object classes are targets - This allows flexibility while maintaining efficient exploration and planning.
> >
> >     For the claim about 60%, it refers to MiraKhor et al., table 1, where #V / #O gives the initially visible percentage, which is 60% for all settings in their settings. We will add this information in the paper along with the visibility percentage of objects in our datasets.
> >
> > So the tradeoff for not performing each goal configuration walkthrough is the prior knowledge of target object classes? Please add the information for 60% visibility setting in the paper.
> >
> >     New Metrics: We will also add a separate table in the appendix with results on how much time was taken to complete the tasks.The distance the agent took to solve the problem is directly proportional to the total actions taken and, hence, not very beneficial in providing any new insight into the method's effectiveness.
> >
> > I did not find the results in the Appendix. Is the paper PDF updated? If no, kindly update the PDF.
> >
> >     Scalability: We currently have results for 2,3,4 rooms with the same number of objects (10) in the MultiRoomR dataset. RoomR and ProcThor dataset results also show this - both datasets have 5 objects but are different in the number of rooms (1-2). This shows how the method behaves with different numbers of rooms with the same number of objects.
> >
> >     We will provide results for larger numbers of objects and add them in the paper (up to 20)
> >
> > I could not find the results in the paper, kindly update the PDF.

---

> ### Author Response · Authors · 2024-11-17
> **Response Part 2**
>
> Response continued
>
> >W4: “The two baselines - PK and PD used in the paper study only the perception efficacy, what about the planning efficacy? Can you replace your planner with some alternatives such as a classical traveling salesman problem (TSP) solver, an optimizer based OR-Tools[7] planner, a greedy planner etc. This will give an insight of how close to the optimal is this planner and how much of an improvement is this method over the heuristic strategies.”
>
> **Ablation on Planning**
> TSP solvers are limited to fully observable domains. The OR methods either assume full observability or (methods like POMCP [7]) are too inefficient for our purpose. The current planner being used (modified PO-UCT) can be turned into  a greedy planner by reducing the look-ahead depth (currently 12) to 1 step. We can run and present results for **different depths (from 1-4)** to show  the importance of look-ahead for planning.
>
> >W5: “As far as I know, ProcThor has multi-room scenarios with up to 5 rooms and about 20 objects. But, the authors have stated in Line 418-420 about ProcThor having - “2 rooms, 5 objects”. Are you sure about this? This begs the question regarding the motivation of the new dataset - Multi RoomR? What was missing in ProcThor? What is the object, receptacle, room type distribution in the new dataset? How do we gauge the complexity of this new dataset, if there are no comparison results with SOTA methods?”
>
> Yes, you are right, the ProcThor dataset itself has multi-room scenarios with more rooms (up to 10). We meant the  **ProcThor Rearrangement dataset** (<https://github.com/allenai/procthor-10k/tree/rearrangement-2022>), which contains multi-room settings with 2 rooms and 5 objects. We will correct this in the paper.
>
> **Motivation for MultiRoomR:** The main motivation for the MultiRoomR dataset is to have more rooms and more objects than existing datasets (RoomR and ProcThorRerrangement), and most importantly, blocked path scenes that do not exist in any dataset.
> For details on the generation of the Novel dataset and room-object distribution, please refer to the **common response**.
>
> [1] Gabriel Sarch, Zhaoyuan Fang, Adam W. Harley, Paul Schydlo, Michael J. Tarr, Saurabh Gupta, and Katerina Fragkiadaki. 2022. TIDEE: Tidying Up Novel Rooms Using Visuo-Semantic Commonsense Priors. In Computer Vision – ECCV 2022.
> [2] Kant, Y.; Ramachandran, A.; Yenamandra, S.; Gilitschenski, I.; Batra, D.; Szot, A.; and Agrawal, H., Housekeep: Tidying Virtual Households using Commonsense Reasoning. In European Conference on Computer Vision, 2022.
> [3] Karan Mirakhor, Sourav Ghosh, Dipanjan Das, and Brojeshwar Bhowmick. Task Planning for Visual Room Rearrangement under Partial Observability. In The Twelfth International Conference on Learning Representations, 2024.
> [4] Karan Mirakhor, Sourav Ghosh, Dipanjan Das, and Brojeshwar Bhowmick.Task Planning for Object Rearrangement in Multi-Room Environments. In Proceedings of the AAAI Conference on Artificial Intelligence, volume 38, pp.10350–10357, 2024.
> [5] Gadre, S.Y., Ehsani, K., Song, S., & Mottaghi, R. (2022). Continuous Scene Representations for Embodied AI. 2022 IEEE/CVF Conference on Computer Vision and Pattern Recognition (CVPR), 14829-14839.
> [6] Brandon Trabucco and Gunnar A Sigurdsson and Robinson Piramuthu and Gaurav S. Sukhatme and Ruslan Salakhutdinov, A Simple Approach for Visual Room Rearrangement: 3D Mapping and Semantic Search, The Eleventh International Conference on Learning Representations, 2023.
> [7] Silver D, Veness J. Monte-Carlo planning in large POMDPs. Advances in neural information processing systems. 2010;23

---

> ### Comment · Reviewer_Jcan · 2024-11-27
>
> Ablation on Planning TSP solvers are limited to fully observable domains. The OR methods either assume full observability or (methods like POMCP [7]) are too inefficient for our purpose. The current planner being used (modified PO-UCT) can be turned into a greedy planner by reducing the look-ahead depth (currently 12) to 1 step. We can run and present results for different depths (from 1-4) to show the importance of look-ahead for planning.
>
> I understand that TSP and OR methods are fully observable in nature. But can they leverage the belief and abstraction module to plan even under partial observability. I mean can the authors decouple the aforementioned modules from their pipeline and make it work with any downstream planner. Also if partial observability is the limitation, how does the proposed method compare to TSP and OR under full observability? I would also like to see the impact of the look-ahead depth on the planning performance.
>
>     Motivation for MultiRoomR: The main motivation for the MultiRoomR dataset is to have more rooms and more objects than existing datasets (RoomR and ProcThorRerrangement), and most importantly, blocked path scenes that do not exist in any dataset. For details on the generation of the Novel dataset and room-object distribution, please refer to the common response.
>
> Nearly 70% of the dataset is composed of only 2 room scenarios, similar to ProcThorRearrangement. Only new addition, the dataset brings to the community is the blocked path scenario.

---

> ### Author Response · Authors · 2024-11-28
> **Response**
>
> >1: “If it is easy to integrate the belief model into the pipeline, I request the authors to kindly show some results for the method with commonsense prior based initial belief v/s random initial belief.”
>
> While it is easy to integrate a common-sense prior when available, we do not have a common-sense sense-prior available for any of the datasets that we have created. The MultiRoomR dataset was created with object placement being random, and hence, the only prior in this case is uniform random distribution.
> There is no easily available prior dataset for the RoomR or the ProcThorRearrangement dataset.In [2], they are learning in a different environment [habitat datasets] and, hence, are not applicable to the RoomR dataset. In [1], they learn an Out of Place detector to detect objects out of place and use common sense prior only for goal locations - where the objects must end up and not where could be in an untidy house. Priors are learned over RoomR objects in [3] and [4], but their code and datasets are not available (we did not get them from the authors after multiple requests). Hence, to test with a prior, we would have to learn one from scratch from the dataset, which is non-trivial, , [3] and [4] (they learn specialized networks for it), and beyond the scope of our work.
>
> >2: “I understand that the method proposed can perform multiple goal configurations. Making the experimental setup for single goal configuration, the comparison study will tell us how the proposed method compares to the existing methods in a traditional rearrangement setting with goal stage walkthrough and shuffle phase rearrangement.”
>
> **Yes**, we can compare our results with existing work on the RoomR dataset (we cannot compare on other datasets as we do not have access to their code or dataset). We have run our system on the RoomR dataset, and we have results from [3] and [4] on the RoomR dataset.
>
> |         Method         | Ours | Mirakhor et al[3]  | Mirakhor et al[4] |
> |:----------------------:|:----:|:-------------------:|:-------------------:|
> |                        |      |      [Table 2]      |      [Table 4]      |
> |                        |      |                     |                     |
> | Scene Success Rate (%) |  49  |          43         |          34         |
>
>
> We can see that our success rate is slightly higher (49 vs. 43 and 34), but this is with the caveat that we use more information.
>
>
> >3: “So the tradeoff for not performing each goal configuration walkthrough is the prior knowledge of target object classes? Please add the information for 60% visibility setting in the paper.”
>
> **Yes**.  Our formulation makes the problem easier by providing the object target classes for the scene being solved but this information helps avoid the need of walkthrough each time. Citation has been added to the paper for 60%.
>
> >4 and 5 - Additional Results
>
> **Table 1** has updated numbers with results on larger number of objects (15 and 20 objects in 3-4 rooms setting). **Table 3** (in appendix A.4) has time information
>
> >6: “I understand that TSP and OR methods are fully observable in nature. But can they leverage the belief and abstraction module to plan even under partial observability. I mean can the authors decouple the aforementioned modules from their pipeline and make it work with any downstream planner. Also if partial observability is the limitation, how does the proposed method compare to TSP and OR under full observability? I would also like to see the impact of the look-ahead depth on the planning performance.”
>
> MCTS over belief states is well-suited for our needs because the search can be interleaved with execution and does not need an exhaustive look-ahead in depth or width. It is possible that it can be replaced with some OR methods that have similar properties, but we could not think of any obvious candidates.  With full observability, the proposed method reduces to MCTS over world states (at the high level). Some solutions to TSP, e.g., simulated annealing, might be suitable and competitive with MCTS. However, we note that partial observability is a fundamental aspect of the problem we address, and comparisons under full observability are not very relevant.
>
> For results at different depth: We have provided results for MCTS depth 1 in **table 3 (Appendix A.4)**. We will have results for **depth 2 and 4 ready for camera-ready version**.
>
> >7: “Nearly 70% of the dataset is composed of only 2 room scenarios, similar to ProcThorRearrangement. Only new addition, the dataset brings to the community is the blocked path scenario.”
>
> Yes, a large percentage of the dataset consists of 2 room settings similar to ProcThorRearrangement, but all of the 2 room settings in our dataset contain 10 objects, whereas ProcThorRearrangement rooms contain 5 objects. Also, the dataset has now been greatly expanded to include more scenes with larger number of rooms as well as larger number of objects[upto 20]. Details in Appendix A.2.

---

> ### Author Response · Authors · 2024-11-28
> **Citations for response**
>
> [1] Gabriel Sarch, Zhaoyuan Fang, Adam W. Harley, Paul Schydlo, Michael J. Tarr, Saurabh Gupta, and Katerina Fragkiadaki. 2022. TIDEE: Tidying Up Novel Rooms Using Visuo-Semantic Commonsense Priors. In Computer Vision – ECCV 2022.
>
> [2] Kant, Y.; Ramachandran, A.; Yenamandra, S.; Gilitschenski, I.; Batra, D.; Szot, A.; and Agrawal, H., Housekeep: Tidying Virtual Households using Commonsense Reasoning. In European Conference on Computer Vision, 2022.
>
> [3] Karan Mirakhor, Sourav Ghosh, Dipanjan Das, and Brojeshwar Bhowmick. Task Planning for Visual Room Rearrangement under Partial Observability. In The Twelfth International Conference on Learning Representations, 2024.
>
> [4] Karan Mirakhor, Sourav Ghosh, Dipanjan Das, and Brojeshwar Bhowmick.Task Planning for Object Rearrangement in Multi-Room Environments. In Proceedings of the AAAI Conference on Artificial Intelligence, volume 38, pp.10350–10357, 2024.

---

### Official Review · Reviewer_9vs1 · 2024-11-04

**Soundness:** 2
**Presentation:** 2
**Contribution:** 2
**Rating:** 3
**Confidence:** 3

**Summary:**

The approach presents a POMDP based hierarchical planning approach for object rearrangement in multi-room settings. As far as I understand, the approach starts with high-level and low-level actions and performs hierarchical POMDP planning. Before the planning, the approach runs a SLAM equivalent  and builds a map of the environment. After, it performs POMDP planning and executes actions. The paper evaluates the approach in AI2THOR multi-room settings.

**Strengths:**

Object rearrangement, i.e., object pick and place tasks are important. The idea that only a subset of objects are visible and the overall setting is a POMDP is important for long-term generalist robots.

The also addresses problems with block paths where it has to re-arrange objects in order to free up the space in order to move. This problem is rarely handled by many approaches as it is a hard problem.

**Weaknesses:**

The paper suffers from the following weaknesses:

- **Lack of clarity**: A few claims in the paper are not clear and unsubstantiated. E.g.,
    - It is unclear what kind of hierarchies are being used in the paper after mentioning it multiple times throughout the paper. The paper mentions about high-level subgoals and low-level actions, however, never defines what constitutes as a high-level subgoal and what actions are low-level actions? The problem definition describes actions but doesn't make the distinction. The paper should clarify this.
    - The paper assumes that whether an action is executed successfully or not is known. How does it ensure this in a POMDP setting where there is a probabilistic observation model.
    - The transition model includes a "PickPlace" action. Is it a single action? If it is does it mean the agent can take this and the object magically transforms to the target location?
    - The approach compares in settings where the agent has to remove obstacles in order to move from one location to another. This is good! However, how is known that picking some action would free up that space? It requires more than a object oriented state space which the paper doesn't clearly explain.
    - It is unclear how the approach collects initial belief of the object location to even build any policies?

- **Lack of Novelty**: The paper claims that this the first approach that does object rearrangement in multiple rooms formulating it as a POMDP and the existing methods assume object locations. However, most of the current approaches such as TAMP (Curtis et al. 2022, Shah et al. 2020) operate in continuous state and action settings. The presented approach operates in discrete state and action spaces making the problem much simples. It is unclear how the approach is any different from any existing POMDP solvers?

References:

Shah, Naman, et al. "Anytime integrated task and motion policies for stochastic environments." 2020 IEEE International Conference on Robotics and Automation (ICRA). IEEE, 2020.

Curtis, Aidan, et al. "Long-horizon manipulation of unknown objects via task and motion planning with estimated affordances." 2022 International Conference on Robotics and Automation (ICRA). IEEE, 2022.

**Questions:**

Please refer to the weaknesses.

---

> ### Author Response · Authors · 2024-11-17
> **Response part 1**
>
> We thank the reviewer for these constructive comments.
>
>
> >W1.1: “It is unclear what kind of hierarchies are being used in the paper after mentioning it multiple times throughout the paper. The paper mentions about high-level subgoals and low-level actions, however, never defines what constitutes as a high-level subgoal and what actions are low-level actions? The problem definition describes actions but doesn't make the distinction. The paper should clarify this.”
>
> **Low-level action and sub-goal clarification:**
> The paper currently mentions in **Line 168** that the low-level actions are picked from $A_{s}$ ($A_s$ is defined in the previous section on **Line 142**). For clarity, we will add the information that these are the low-level actions in line 142. The actions output by the abstract planner are the sub-goals for the low level control. The relation between them is mentioned in lines 380 onwards. We will add this  to the definition of  abstract POMDP for further clarity.
>
> >W1.2: “The paper assumes that whether an action is executed successfully or not is known. How does it ensure this in a POMDP setting where there is a probabilistic observation model.”
>
> **This is an optimistic model of planning** that assumes the abstract actions will succeed and plans accordingly. The failures are handled through replanning at execution time. We get the information about action failure (action success/failure part of the observation is deterministic; object detection is probabilistic) from the environment and update the current state accordingly - since we re-plan at each step, the planning now starts with the state where the action has failed and hence our system can solve problems where actions can fail.
>
> >W1.3: “The transition model includes a "PickPlace" action. Is it a single action? If it is does it mean the agent can take this and the object magically transforms to the target location?”
>
> **PickPlace details:**
> **No**, the agent cannot magically transform the object to the target location. The PickPlace is an abstract action defined for the abstract POMDP planner. **There exists a low-level policy that takes this single PickPlace action** (output by the planner) and executes low-level actions to achieve it. The details are provided from line 390 onwards on how the policy uses the information from the abstract action to come up with a sequence of low-level actions to achieve the pick, move, and place.
>
> >W1.4: “The approach compares in settings where the agent has to remove obstacles in order to move from one location to another. This is good! However, how is known that picking some action would free up that space? It requires more than a object oriented state space which the paper doesn't clearly explain.”
>
> The state of an object is represented with the following information: (loc_i, pick_i, place_locs, is_held, at_goal). When the path to an object obj_1 is blocked, there will be no pick_i locations present (hence, the planner will not have picking of obj_1 as part of the plan). When an object obj_2 is picked up, **the transition model updates the state that this object is no longer occupying this location**, the state abstraction system's sampler can now sample location pick_i from where obj_1 can be picked up, thereby allowing the planner to have picking of obj_1 as part of the plan. (more details in section 4.3, generating abstract state).
>
> >W1.5: “It is unclear how the approach collects initial belief of the object location to even build any policies?”
>
> **Initial belief and behaviour:**
> **We start with uniform belief over the entire state**. Hence, in the very beginning, the planner is operating with no information, but as it plans, executes, and receives observation, its belief about object location keeps improving (at any given step, if an object is detected, then our belief about the object's viewed location is greatly increased. If an object is not seen, then the belief that the object exists in any of the visible locations is reduced). (Algorithm 2, line 7 shows this update. P(z_i | s_ij) is detailed in Appendix A.1.2)

---

> > ### Comment · Reviewer_9vs1 · 2024-11-25
> >
> > Thank you, authors, for your response.
> >
> > From the response:
> >
> > - It is still unclear what the high-level actions are and what makes something a subgoal.
> > - The authors seem to have missed the point. The question is: At the execution time, given the state is unknown and only access to observation is available, how are actions determined to have failed? A failure can only be known if the actual state of the system is known - making it an MDP and not a POMDP.
> > - The third point raises a few more questions (and sorry for raising them now): How does A* work for a POMDP? How is RL model trained in a POMDP?
> > - Thanks for clarifying the moving of different objects. However, it is a strong assumption. Especially for an object-centric observation space where operating on one object would change the state of another object, and we would know in a POMDP setting. Mathematically speaking, in reality, $pick_i$ is still there. But only it is not accessible. TAMP papers make this clear. I would suggest that authors refer to them.
> > - (From the second part) Can you clarify how a single-room setting is different than a multi-room setting in POMDPs? Here, My original point refers to older POMDP solvers such as POMCP. Especially given a discrete state and action spaces.
> >
> >
> > Because of all these issues, I would like to keep my current score as it is. Looking forward to response from the authors.

---

> ### Author Response · Authors · 2024-11-17
> **Response part 2**
>
> Response contd.
>
> >W2: “The paper claims that this the first approach that does object rearrangement in multiple rooms formulating it as a POMDP and the existing methods assume object locations. However, most of the current approaches such as TAMP (Curtis et al. 2022, Shah et al. 2020) operate in continuous state and action settings. The presented approach operates in discrete state and action spaces making the problem much simples. It is unclear how the approach is any different from any existing POMDP solvers?”
>
> **Comparison to other works in the area:**
> Among the mentioned papers (both [1] and [2]) operate in a fully observable world - they have no uncertainty over object locations and do not need to account for that in their planning, which is major part of the problem we are solving - planning to reduce this uncertainty and achieve the goal efficiently. Yes, our action space is discrete, but the state space is not. The input to our system is RGBD image and goal locations which are in R^3 (continuous 3D coordinates). We discretize the world using our perception system to make the problem solvable by the planner.
>
> Existing POMDP Planners are **limited to solving object search** (Zheng et al, 2022[3]; Zheng et al. [4], 2023) or **object rearrangement in a small region** (Caelen et al [5];  Pajarinen et al [6]). Our work is the first to apply **POMDP to rearrangement in multi-room environments**. It is made possible by extending the object-oriented formulation defined in Zheng et al, 2022 to rearrangement tasks and abstracting this formulation to make it efficient in solving rearrangement tasks in multi-room environments.
>
>
> [1] Shah, Naman, et al. "Anytime integrated task and motion policies for stochastic environments." 2020 IEEE International Conference on Robotics and Automation (ICRA). IEEE, 2020.
>
> [2] Curtis, Aidan, et al. "Long-horizon manipulation of unknown objects via task and motion planning with estimated affordances." 2022 International Conference on Robotics and Automation (ICRA). IEEE, 2022.
>
> [3] Kaiyu Zheng, Rohan Chitnis, Yoonchang Sung, George Konidaris, and Stefanie Tellex. Towards optimal correlational object search. In 2022 International Conference on Robotics and Automation (ICRA), pp. 7313–7319. IEEE, 2022.
>
> [4] Kaiyu Zheng, Anirudha Paul, and Stefanie Tellex. A System for generalized 3d multi-object search.In 2023 IEEE International Conference on Robotics and Automation (ICRA), pp. 1638–1644.IEEE, 2023
>
> [5] Caelan Reed Garrett, Chris Paxton, Tomas Lozano-Perez, Leslie Pack Kaelbling, and Dieter Fox.  Online replanning in belief space for partially observable task and motion problems. In 2020 IEEE International Conference on Robotics and Automation (ICRA), pp. 5678–5684. IEEE, 2020b
>
> [6] Joni Pajarinen, Jens Lundell, and Ville Kyrki. Pomdp manipulation planning under object composition uncertainty. arXiv preprint arXiv:2010.13565, 2020.

---

> ### Author Response · Authors · 2024-11-26
> **Response Part 1**
>
> We thank the reviewer for these constructive comments.
>
> >Q1: It is still unclear what the high-level actions are and what makes something a subgoal.
>
> The high-level actions are (action space of the Abstract POMDP Planner):
> 1. **MoveAB**  - move action that moves the agent from location A to location B
> 2. **Rotate_angle** -  The rotate action rotates the agent to a given angle
> 3. **PickPlace** - The PickPlace action picks Object_i from the current position of the robot and places it at the given goalloc
>
> A **subgoal is an instantiated high-level action**. For example, **MoveAB((5,5), (10,10))** is a subgoal instantiated from the high-level action MoveAB. (An instantiated action sequence is what the abstract planner outputs after planning). The low-level policies are initialized using this information - A* is initialized with the starting position (5,5) and goal position (10,10) to find a sequence of low-level actions to move between these locations.
>
> > Q2: “The authors seem to have missed the point. The question is: At the execution time, given the state is unknown and only access to observation is available, how are actions determined to have failed? A failure can only be known if the actual state of the system is known - making it an MDP and not a POMDP”
>
> By definition, in a POMDP, we do not know the full state of the world and hence maintain a belief over all possible states. In our case, the state is represented by (s_r, s_objects) - the state of the robot (s_r), and state of the objects (s_objects)
>
> Action success/failure information is part of s_r (along with robot position (x,y, pitch, yaw)). This part of the state is fully known at any given time from the observation z_r, which is deterministic (the simulator gives agent position along with success/failure information at each step). **What is NOT known is the s_objects - the locations of the objects in the world**. For this, we get only PARTIAL information about the world - in the form of RGB and depth images (which is a first-person view, so only a small part of the environment is visible at any given time).
> Our perception system converts this to an observation based on detection (which can also fail), and we update our belief based on this.
>
> >Q3: “The third point raises a few more questions (and sorry for raising them now): How does A* work for a POMDP? How is RL model trained in a POMDP?”
>
> Both A* and RL are policies that work at a low level and are not affected by the partial knowledge of the world.
>
> We train 2 RL policies - Pick RL and Place RL. They take the RGB and Depth images as input along with the object information (object name for Pick policy and goal location for Place policy)
> Both of these are meant to interact with a single object at any given time and do not use the full state (which is unknown due to partial observability).
>
> For example, **PickPlace (book, (11,11) )** and the agent’s position is at (5,5) currently. This means we must pick the object book from the current location and place it at (11,11).
> How this gets broken down and solved is as follows:
>
> 1. First, the Pick RL policy is called to Pick(Book) in its vicinity only (note that if there is no book nearby, then the pick will simply fail).  In case of failure, the abstract POMDP planner is expected to instantiate pick from a different location to be able to pick this object.
> 2. Once pick has happened -> Recall that we have a 2D obstacle map of the world (of only stationary objects). We compute the nearest position to (11,11), that is free to move to based on this obstacle map. Let’s say this is (10,10). A* is initialized - to move from (5,5) to (10,10). A* finds a sequence of actions to move from (5,5) to (10,10) while avoiding these obstacles in the 2D map.
> ( (It plans a path as if there is nothing on the way - if our agent finds something on the way later (in a blocked goal setting), the abstract POMDP planner is expected to output a different high-level goal during re-planning - which happens after each low-level action has been taken))
> 3. Place RL: Once we are at (10,10), the place policy takes over and tries to place the book at (11,11).
>
> Note that, in all cases, we only used known information for the low-level policies—the location of the obstacles for A*. The RL policies also only care about the object of interest and not the state of any other objects.

---

> ### Author Response · Authors · 2024-11-26
> **Response Part 2**
>
> > Q4. Thanks for clarifying the moving of different objects. However, it is a strong assumption. Especially for an object-centric observation space where operating on one object would change the state of another object, and we would know in a POMDP setting. Mathematically speaking, in reality, $pick_i$ is still there. But only it is not accessible. TAMP papers make this clear. I would suggest that authors refer to them.
>
> It is true that in the real world, an object may be impacted while another one is moved. However, our high-level planner still makes this optimistic independence assumption and produces a plan. If, in fact, there are unintended interactions during the execution, e.g., another object falls down while the first object is picked up, the detector is (hopefully) going to detect this, and the planner replans. This approach reflects the optimistic planning people typically engage in rather than getting bogged down in modeling and considering all possible interactions at the planning time.
>
> >Q5:  “Can you clarify how a single-room setting is different than a multi-room setting in POMDPs? Here, My original point refers to older POMDP solvers such as POMCP. Especially given a discrete state and action spaces”
>
> Theoretically, a multi-room setting is no different from a single-room setting for a POMDP (assuming you had infinite compute resources, you could solve them both with the basic POMDP formulation).
>
> Practically, there are two issues -
>
> 1. **State space increase**: Let’s consider a room of size 5*5. If we have 10 objects, the state space, where any object can be in any location, is 25^10 ~ 10^14. But if we have 4 rooms - then we have a house of size 10*10, and state space size goes up to 100^10 ~ 10^20. As we can see, the state space of a multi-room problem is orders of magnitude larger than a single-room setting.
>
> As we further increase the size of the room and the number of objects, the state space increases exponentially. Algorithms like POMCP[1] can handle this size of state space but need a very large number of MCTS simulations (10^4 to 10^5 (figure 2, POMCP [1])), whereas we use only 500 simulations. This is possible due to our object-oriented belief update (POMCP[1] uses a particle-based approximate belief update, which leads to inaccuracies and needs more simulations, whereas we can do a full Bayes update for each object independently). Hence, our first contribution of extension of OOPOMDP to rearrangement POMDP makes it possible to handle multi-room scenarios.
>
> 2. Second, **The percentage of the environment visible becomes much smaller** - we can see only 10% of the environment in any given multi-room setting, whereas we can potentially see 50% of the world in single-room settings. This implies we need to perform a lot more actions to interact in the full world (we need to go to each room and explore it).
>
> Hence, the depth of search required is very high in a rearrangement task - upto 1000 steps of low-level actions (with 4 rooms and 10 objects) to solve a single problem, making each simulation very expensive and also unlikely to find a solution. That is where our second contribution of abstract OO-POMDP becomes extremely useful - with the provided abstraction, the depth of the plan scales linearly with the number of objects and does not depend on the size of the room - thereby speeding up the planning process considerably (planning depth is about 3-4x the number of objects - hence a depth of ~40 for the abstract planner).
>
> Hence, our contribution of the Abstract OO-POMDP for rearrangement is a vital factor in making a POMDP solution viable in a multi-room setting.
>
> [1] Silver D, Veness J. Monte-Carlo planning in large POMDPs. Advances in neural information processing systems. 2010;23

---

### Official Review · Reviewer_BB3F · 2024-11-04

**Soundness:** 3
**Presentation:** 2
**Contribution:** 3
**Rating:** 6
**Confidence:** 3

**Summary:**

The paper addresses a variation of a challenging POMDP setting, an object rearrangement task over multiple rooms, with imperfect object detection. The authors introduce a hierarchical approach to a solver, with planning over a computed abstract state, and trained low-level policies to execute high-level plans. The work tests this method on existing object rearrangement tasks, and introduces a dataset of additional, harder tasks, based on the AI2Thor simulator. The authors also provide experimental evaluations of their solver on these domains.

**Strengths:**

The paper has several strengths, with a new method proposed for a relatively novel setting, with experimental results and a contribution to the field in the form of a dataset of tasks. On originality, the paper tackles a more challenging extension of the multi-room rearrangement problem, i.e. adding imperfect detection and integrated decision making. They additionally do not assume perfect navigation, motion planning or manipulation, instead having trained or computed low-level policies. The work contributed is mostly of high quality (with some reservations described below), with experimental results that demonstrate their method works in harder domains (that were also contributed and represent a meaningful improvement over the current domains). The lack of assuming perfect object detection and perfect low-level control renders this contribution significant to the field.

**Weaknesses:**

However, there are some concerns with the paper. Firstly, the paper's presentation and clarity desperately needs to improve. Specifically around sections 3, 4.1, and 4.2, there are many open questions, missing details and unclear statements around the task definition and setup. For instance, how the agent works with the 2D and 3D maps, how does it learn information about the receptacles at the beginning, or how the observations are discretized before being passed to the belief update is unclear. These can impact the evaluation of the results, making the environment and task easier than initially understood, and could potentially rely on unrealistic assumptions, for which is there is also no discussion on. I have listed a lot of questions I had around this section later on, and I can only be confident in the results provided, if the authors can improve the clarity of the paper here.

Additionally, there are other concerns with the results. The authors claim in Section 6 that existing baselines differ from other work in key aspects. I think a detailed comparison between your domain and method, versus other selected variants of the task and accompanying methods would significantly improve  the quality of the paper's results.  At the moment, all the authors say is:

> The primary distinction lies in the prior knowledge available to our system: we are given information
about the classes of objects to be moved, whereas other systems operate without this advantage.
but do not cite other systems or prior work that studies the other settings.

Further, they claim that:

> In particular, while existing systems report initial visibility of approximately 60% of target objects
at the outset of their tasks, our scenarios present a more demanding exploration challenge. Only
about 20% of the objects are initially visible in our problem settings, necessitating more extensive
and strategic exploration.

but again, lack a reference for this information. Additionally, from a brief overview, it appears that [Mirakhor et al. 2024](https://ojs.aaai.org/index.php/AAAI/article/view/28902) (note, it was published before the period ICLR allows for concurrent work) is a relevant comparison, as they operate in similar conditions (i.e. multiple rooms, rearrangement task, with similar setups such as swap and blocked goal cases) and provide similar contributions (i.e. novel planner and new dataset). I want to see how the proposed method and provided datasets compare, to contextualize the effectiveness of this approach.

Lastly, the authors mention this (lines 64, 508), but do not elaborate on this in the Limitations section. The proposed method requires a factored object-oriented state representation, and the proposed belief update and abstraction method rely on this fact as well (see enumerations over objects in Algorithm 2 and "Generating Abstract State"). This assumption is fairly strong, and presents a stumbling block in environments where object classes might not be fully known, i.e. imagine you see an unknown object that's blocking the goal location for a known object. I'm not suggesting that the proposed method needs to be able to handle such cases, but a fuller discussion of limitations needs to go beyond just the independence assumption and include ideas of how this might be relaxed in the future or how other methods in literature handle such cases.

My suggestion is to combine and extend the limitations and comparison to existing baselines section and address both weaknesses together. I would be willing to increase my score if these concerns around the clarity of the paper, the discussion around baselines and the discussion of limitations of the method were improved.

**Questions:**

Questions:
- it is not clear how the agent generates the 2D map. The paper says it discretizes the world into grids of size 0.25m, but how is that done? Does the environment provide it? If not, how is it computed from the sequences of observations during the exploration phase? It's even less clear how the agent generates the 3D map mentioned in line 154.
- The setup of the task is unclear. Is the first phase of exploration (where the agent "traverses the world" and "gains location information about the receptacles") something that the agent has to to plan how to do and output a sequence of actions? Or is this information provided by the environment? How does the agent know what the type of the object is during this phase? From the previous paragraph, the environment simply outputs the RGBD image of the current view from the agent POV, the location and if the action was successful. If the agent does have to do this, you must include more detail about how this works, and how it interacts with the planning system provided.
-  The definition of the abstract POMDP is not clear. What change does the 'object independence' assumption make to the mathematical formulation of the OO-POMDP provided above? You provide some detail in the observation model bullet point but you reference "conditional independence" here and "object independence" above, so it's not clear to me what the relationship is between this and the above's "abstract" nature. My suggestion is to describe the overall system first (like the initial paragraph of section 4.3), since that provides much needed context to understand your formalism. Alternatively, I would try and make the abstraction system clearer when you define the abstract POMDP. The current presentation is very confusing.
- How are the object locations in the ground truth image observation discretized to the 2D map? is this done by the environment or the perception system? The understanding is the perception system just runs object detection and grabs the 3D location via the depth map.

Nits (not affecting score):
- typo on line 87 in caption (sawp -> swap)
- typo on line 354 (the OO-POMDP planner,uses Partially)
- typo in line 162, the title of section 4 should have the full form of the acronym, HIERARCHICAL OBJECT ORIENTED POMDP (HOO-POMDP), and should have a space between oriented and the parenthesis.
- formatting of line 230 is wrong (should be in latex math mode, something like: [$cost = -1 \times N_{a}$] where $N_{a}$ is the number of required actions).
- typo in line 248, space should be there between z and This
- the section in lines 196 to 201 are really difficult to read because of how compressed the math definitions are. it would be good to expand them to be easier to read.
- typo in line 212 (null is malformed)
- no need to redefine the acronym in line 316
- space around hypen in line 323
- randomly repeated twice in line 369
- A* is formatted incorrectly in line 380.
- missing period on line 452
- minor, but please bold the best result in the results Table 1, or otherwise easily indicate which performed the best.

---

> ### Author Response · Authors · 2024-11-17
> **Response to reviewer questions**
>
> We thank the reviewer for these constructive comments.
>
>
> >“Q1: it is not clear how the agent generates the 2D map. The paper says it discretizes the world into grids of size 0.25m, but how is that done? Does the environment provide it? If not, how is it computed from the sequences of observations during the exploration phase? It's even less clear how the agent generates the 3D map mentioned in line 154.
>
> >Q2: The setup of the task is unclear. Is the first phase of exploration (where the agent "traverses the world" and "gains location information about the receptacles") something that the agent has to to plan how to do and output a sequence of actions?...  If the agent does have to do this, you must include more detail about how this works, and how it interacts with the planning system provided.”
>
>
> **Overall Task setup and map building:**
> Rearrangement is done in **2 phases**. **Walkthrough** phase and **rearrange** phase. The walkthrough phase is meant to get information about stationary objects. The 2D occupancy map is generated in this phase, as well as the corresponding 3D Map. First, we get the size of the house(width and length) information from the environment. Then, we uniformly sample points in the environment then take steps to reach these locations (if possible - some might be blocked). This simple algorithm ensures we go all around the house and see every part of it. At each of the steps involved in reaching these locations, we receive the RGB and Depth image from the environment. Using this, **we create a 3D point cloud at each step and combine them all together to get the overall 3D point cloud** of the house with stationary objects. We then **discretize this point cloud into 3D map voxels of size 0.25m, we further flatten this 3D map into a 2D map** (location in the 2D map is occupied if there exists a point at that 2D location at any height in the 3D map - after flattening, voxels becomes grid blocks of size 0.25m a side). While doing this traversal, we also get information about the receptacles by detector on the RGB images we receive during this traversal. This ends the walkthrough phase. (this walkthrough process is similar to other works solving the rearrangement problem [1][2], except that it needs to be done only once for any house configuration of stationary objects - walls, doors, tables, etc.). Then, objects are placed at random locations (done using AI2Thor environment reinitialization). This is when the rearrangement phase begins, with the planner taking the following as **input - the map generated in the walkthrough phase, the set of object classes to move, and their goal locations**.
>
> This is not explicitly mentioned in the paper because it is standard practice for rearrangement tasks (Mirakhor et al [4]), but we will add a summary for completeness in the task setup section.
>
> > Q3: “The definition of the abstract POMDP is not clear. What change does the 'object independence' assumption make to the mathematical formulation of the OO-POMDP provided above? … . My suggestion is to describe the overall system first (like the initial paragraph of section 4.3), since that provides much needed context to understand your formalism. Alternatively, I would try and make the abstraction system clearer when you define the abstract POMDP. The current presentation is very confusing.”
>
> **The object independence assumption** : defined as “the observation and belief of any object do not depend on any other object”,  can be formally stated as **(P(z_i| s_j,z_j, s_i ) = P(z_i | s_i) for all j != i)**. Observation z_i  is independent of the states and observations of other objects, given its own state s_i. Similarly, we also assume **P(s’_i|s_i,s_j,a) = P(s’_i|s_i,a)** when j != i, i.e., the next state of object i only depends on its own previous state and the action.  **These two assumptions help us go from equation 1 to 3, as well as perform belief updates independently for each object (algorithm 2)**. We will add this clarification in the paper to make it clearer.
>
> Thank you for the suggestion about moving sections. To improve clarity, we will move the initial paragraph of section 4.3 to the beginning of section 4 and then describe the rest of the system.
>
> > Q4:“How are the object locations in the ground truth image observation discretized to the 2D map? is this done by the environment or the perception system? The understanding is the perception system just runs object detection and grabs the 3D location via the depth map.”
>
> Yes, we get the local 3D coordinates  (x,y,z) from the depth map w.r.t. Agent. We convert this to global coordinates using the agent’s global position. We then drop the third dimension to map it to a grid in the 2D map.
>
> >Typos: Thank you for pointing out the typos, we will fix them

---

> ### Author Response · Authors · 2024-11-17
> **Responses to weaknesses**
>
> Response continued
>
> >W2: "Additionally, there are other concerns with the results. The authors claim in Section 6 that existing baselines differ from other work in key aspects. I think a detailed comparison between your domain and method, versus other selected variants of the task and accompanying methods would significantly improve the quality of the paper's results. … ”
>
> **Missed citations:** Thank you for pointing it out. We will add citations for both of these. For the claim about other papers needing extra information, it is every other system that does rearrangement, we will cite the most relevant ones in the paper (Sarch et al.[1], Kant et al.[2] and Mirakhor et al. [3,4]).   **For the claim of 60%, it is taken from MiraKhor et al.[4], table 1** (where #V/#O gives the initially visible percentage, which is 60% for their settings).
>
> >W3: “I want to see how the proposed method and provided datasets compare, to contextualize the effectiveness of this approach.”
>
> We were unable to get the code or their dataset from the authors (after multiple requests) of Mirakhor et al [4] and hence unable to make a direct comparison.
>
> >W4: “Lastly, the authors mention this (lines 64, 508), but do not elaborate on this in the Limitations section. The proposed method requires a factored object-oriented state representation, and the proposed belief update and abstraction method rely on this fact as well. …  a fuller discussion of limitations needs to go beyond just the independence assumption and include ideas of how this might be relaxed in the future or how other methods in literature handle such cases.”
>
> Indeed, we cannot currently handle cases where an unknown object is in the way. A simple way to handle this is to group all unknown objects into one class, and **whenever an unknown class object blocks a path, we place it into an empty receptacle** (similar to how we handle swap/blocked goal cases). We will add this in the limitations section.
>
> [1] Gabriel Sarch, Zhaoyuan Fang, Adam W. Harley, Paul Schydlo, Michael J. Tarr, Saurabh Gupta, and Katerina Fragkiadaki. 2022. TIDEE: Tidying Up Novel Rooms Using Visuo-Semantic Commonsense Priors. In Computer Vision – ECCV 2022.
>
> [2] Kant, Y.; Ramachandran, A.; Yenamandra, S.; Gilitschenski, I.; Batra, D.; Szot, A.; and Agrawal, H., Housekeep: Tidying Virtual Households using Commonsense Reasoning. In European Conference on Computer Vision, 2022.
>
> [3] Karan Mirakhor, Sourav Ghosh, Dipanjan Das, and Brojeshwar Bhowmick. Task Planning for Visual Room Rearrangement under Partial Observability. In The Twelfth International Conference on Learning Representations, 2024.
>
> [4] Karan Mirakhor, Sourav Ghosh, Dipanjan Das, and Brojeshwar Bhowmick.Task Planning for Object Rearrangement in Multi-Room Environments. In Proceedings of the AAAI Conference on Artificial Intelligence, volume 38, pp.10350–10357, 2024.

---

> > ### Comment · Reviewer_BB3F · 2024-11-25
> >
> > Thank you for the clarifications.
> >
> > Going over my main concerns with the paper, I will evaluate the author's rebuttal and provide an updated review statement.
> >
> > 1. Clarity. The authors provide additional detail in the rebuttal and update the paper. I appreciate these changes, and I think it's a stronger paper for it. I still maintain that this paragraph (Section 3, subheading "Challenge") is unclear:
> >
> > > First, the agent traverses the world, gains location information about all receptacles,
> > and stores it in a list R = {ri
> > , i = 1, ..k}, representing their centroids. The agent also builds a 2D
> > map (M2D) of the world for navigation during rearrangement. This is done by discretizing the world
> > into grids of size 0.25m. The agent also builds a 3D map (M3D) and stores it. Then, a set of objects
> > are placed at random locations in the environments. The agent is put back into the environment at
> > a random location using the startloc action.
> >
> > and it would be better if the authors modified this part of the response and updated that paragraph.
> >
> > > Overall Task setup and map building: Rearrangement is done in 2 phases. Walkthrough phase and rearrange phase. The walkthrough phase is meant to get information about stationary objects. The 2D occupancy map is generated in this phase, as well as the corresponding 3D Map. First, we get the size of the house(width and length) information from the environment. Then, we uniformly sample points in the environment then take steps to reach these locations (if possible - some might be blocked). This simple algorithm ensures we go all around the house and see every part of it. At each of the steps involved in reaching these locations, we receive the RGB and Depth image from the environment. Using this, we create a 3D point cloud at each step and combine them all together to get the overall 3D point cloud of the house with stationary objects. We then discretize this point cloud into 3D map voxels of size 0.25m, we further flatten this 3D map into a 2D map (location in the 2D map is occupied if there exists a point at that 2D location at any height in the 3D map - after flattening, voxels becomes grid blocks of size 0.25m a side). While doing this traversal, we also get information about the receptacles by detector on the RGB images we receive during this traversal. This ends the walkthrough phase. (this walkthrough process is similar to other works solving the rearrangement problem [1][2], except that it needs to be done only once for any house configuration of stationary objects - walls, doors, tables, etc.). Then, objects are placed at random locations (done using AI2Thor environment reinitialization). This is when the rearrangement phase begins, with the planner taking the following as input - the map generated in the walkthrough phase, the set of object classes to move, and their goal locations.
> >
> > The response is a much clearer detailing of the actual setup. While the authors suggest they have left this out from the paper because it is standard practice, I believe it hurts the readability of the paper.
> >
> > Additionally, thanks for fixing the nits, but I did notice many more where there's are missing periods at the ends of many sentences, in line 227, 230, 331, 236, 264 (in the caption), etc. I recommend the authors do a thorough copy-edit and grammatical check of the paper before a camera-ready version--whether for this conference or another.
> >
> > 2. Concern around baselines.
> >     a. Missing citations for claims. Thanks for providing the citations and improving the discussion. This satisfies my critique.
> >     b. Comparison to Mirakhor et. al. Ah, it is unfortunate you were unable to access the code or the datasets after reaching out. I will not hold that against this work. Upon a brief check, I was also unable to locate the code for the paper, including released supplementary material.
> >
> > 3. Discussion of limitations. The authors address the example I provided, which helps flesh out the limitations sections, but that is not sufficient for addressing the critique.
> >
> > > This assumption is fairly strong, and presents a stumbling block in environments where object classes might not be fully known [...]
> >
> > I would like the authors to expand upon their analysis here. What could one do if the state wasn't factored? How would they imagine future work (I note that Section 7 is titled 'Conclusion and Future Work' but does not mention any future work).
> >
> > Regardless, the proposed updates to the paper in the author response do help increase the clarity of the paper somewhat and I am increasing my score to reflect this. However, there are still improvements that can be made in this direction.
> >
> > Note: that the PDF has not been updated to reflect these changes. I would expect the authors ensure an updated copy is present soon during the rebuttal period.

---

> > > ### Author Response · Authors · 2024-11-26
> > >
> > > We thank the reviewer for these constructive comments.
> > >
> > > > “Discussion of limitations. The authors address the example I provided, which helps flesh out the limitations sections, but that is not sufficient for addressing the critique.  “This assumption is fairly strong, and presents a stumbling block in environments where object classes might not be fully known [...]” I would like the authors to expand upon their analysis here. What could one do if the state wasn't factored? How would they imagine future work (I note that Section 7 is titled 'Conclusion and Future Work' but does not mention any future work).”
> > >
> > > **Limitations:**
> > >
> > > Yes, currently, we cannot handle an unknown class of objects. We could potentially handle them by categorizing all of the known object types into a single ‘unknown’ class. The difficult part, however, is to plan to find an empty space to move the unknown object to. In the worst case, this could lead to complicated packing problems which are NP-hard, but assuming that the space is relatively free, it can be handled with a little more additional search.
> > >
> > > **Future Work**:
> > >
> > > One of the ways to expand the scope is to relax the assumption of object independence partially. We can allow objects to be dependent on a small number of objects (e.g., objects in their close vicinity). Belief updates can now consider a small set of objects at any time. This relaxation helps maintain the efficient belief update while accounting for more real-world situations such as object-object interaction. Another potential future work is to handle stacking of objects and more cramped spaces, where more careful reasoning about object interactions is needed to plan the actions and order them appropriately.
> > >
> > > We will add a summary of the above in the paper and fix the nits. We will upload an updated PDF with all of the changes discussed in the rebuttal period soon.

---

### Official Review · Reviewer_fzvQ · 2024-11-04

**Soundness:** 2
**Presentation:** 2
**Contribution:** 2
**Rating:** 6
**Confidence:** 3

**Summary:**

This paper proposes a framework for the multi-object rearrangement problem within a Partially Observable Markov Decision Process (POMDP) setting. The authors introduce a hierarchical, object-oriented POMDP framework that utilizes a high-level planner to generate sub-goals and deploys low-level policies to accomplish these sub-goals effectively. To benchmark their approach, the authors present a new dataset, “Multi RoomR,” designed to address more complex scenarios. This dataset includes a larger number of objects (10 objects) and more extensive environments (2-4 rooms), providing a more challenging testbed. The authors evaluate different variants of their method on this new dataset and two existing benchmarks, demonstrating that their framework achieves performance comparable to the method with perfect knowledge, even under partial observability constraints.

**Strengths:**

1. The authors demonstrated that their framework in a partial observability setting achieves results comparable to those obtained with perfect knowledge.
2. The new dataset introduces more complex scenarios, which is evident from the performance gap. However, additional details about these scenarios would strengthen their contribution, and I suggest they provide further explanation, perhaps in the appendix, to clarify the dataset’s design and the specific challenges it presents.

**Weaknesses:**

1. I find the work is incremental with limited novelty “Zheng et al. (2023) and Zheng et al. (2022) extend this formulation to perform object search in 3D environments. However, they are all limited to the task of object search and do not include any tasks that require rearrangement. In our work, we build on their formulation of object-oriented POMDP and extend it to include rearrangement actions and their corresponding belief updates.” (lines 128-132). Despite, I find you have added a hierarchical POMDP planning as well, but I find it already in the literature, for example [1]. Adding a more distinct methodological advancement or exploring further applications beyond rearrangement might strengthen the impact of this work.

2. The paper lacks details about the newly introduced dataset, which is a key part of the stated contributions in the introduction. For instance, specifics on the types of objects included and the rationale behind their selection are missing. Additionally, there’s little information on how the scenarios were designed—such as the criteria for object placement, room configuration, or how these factors contribute to the complexity of the rearrangement tasks. Providing this information, perhaps in the appendix or a dedicated section, would give readers better insight into the dataset's structure and its intended challenges, thus strengthening the contribution.

3. Their method is evaluated only against variants of itself, lacking comparisons with other baseline approaches. This makes it difficult to assess the true advantages of their approach. I would expect, at a minimum, an ablation study that removes the object-oriented hierarchical planning component to demonstrate its effectiveness compared to flat planning methods.

**Questions:**

**Minor Improvements (Not considered in the score)**

1. BeliefUpdate instead of UpdateBelief function → line 10 in Algorithm 1

**Questions:**

1. Could you clarify your statement at the end of Section 1, where you describe the system as "an end-to-end planning system"? My understanding is that the detection model and low-level policies are trained independently, suggesting a modular rather than fully end-to-end approach.
2. What is the last row of Table1?

---

> ### Author Response · Authors · 2024-11-17
>
> We thank the reviewer for these constructive comments.
>
> >Q1:  Could you clarify your statement at the end of Section 1, where you describe the system as "an end-to-end planning system"? My understanding is that the detection model and low-level policies are trained independently, suggesting a modular rather than fully end-to-end approach.
>
> This is a valid point that it is not an end-to-end trained system. We will use the terminology of modular system in the paper.
>
> >Q2: “What is the last row of Table 1?”
>
>  The last row of the table is our system run in the Multi-room setting with **10 objects for 3-4 room** settings (the previous two rows are for 1-2 room settings). We will remove that horizontal line to avoid any confusion.
>
> >W1:  “I find the work is incremental with limited novelty… . Despite, I find you have added a hierarchical POMDP planning as well, but I find it already in the literature, for example [1]. Adding a more distinct methodological advancement or exploring further applications beyond rearrangement might strengthen the impact of this work.”
>
> **Novelty 1:**  Extend OO-POMDP originally designed for object search to rearrangement tasks. (state abstraction by factoring state based on objects)\
> **Novelty 2:** We further extend this Rearrangement OO-POMDP to a hierarchical planning setting (through action abstraction).
> We will add this clarification about the exact novelty in the paper.
>
> >W2:  “The paper lacks details about the newly introduced dataset, which is a key part of the stated contributions in the introduction.… Providing this information, perhaps in the appendix or a dedicated section, would give readers better insight into the dataset's structure and its intended challenges, thus strengthening the contribution.”
>
> Please refer to the **common response** for details about the dataset.
>
> >W3:  “Their method is evaluated only against variants of itself. I would expect, at a minimum, an ablation study that removes the object-oriented hierarchical planning component”
>
> We provide results for the flat object-oriented POMDP here(ablation on hierarchy). **OURS represents our method, OURS-HP represents the approach without the hierarchical planning** in the table below which clearly shows the importance of hierarchy  and abstraction (The planner directly outputs low-level actions). Unfortunately object-oriented representation and the independent belief update are too critical to the method in that without them even the simplest of problems are not going to be solved in a reasonable time.  We will add this in the appendix.
>
> | Dataset         | Objs | #BG | #Swap | #BP | #RM | OURS (SS) | OURS (OSR) | OURS (TA) | OURS-HP (SS) | OURS-HP (OSR) | OURS-HP (TA) |
> |------------------|------|------|--------|------|------|-------------|-------------|-------------|----------------|----------------|----------------|
> | **RoomR**        | 5    | 1    | 0      | 0    | 1    | **49**          | **71**          | **211**         | 13             | 33             | 302            |
> |                  | 2    | 2    | 2      | 1    | 1    | **39**          | **61**          | **289**         | 8              | 27             | 392            |
> | **Proc**         | 5    | 1    | 0      | 0    | 2    | **46**          | **68**          | **352**         | 9              | 29             | 410            |
> |                  | 2    | 2    | 1      | 2    | 1    | **31**          | **53**          | **398**         | 4              | 19             | 565            |
> | **Multi RoomR**  | 10   | 1    | 0      | 0    | 2    | **32**          | **65**          | **710**         | 5              | 25             | 1029           |
> |                  | 10    | 2    | 1      | 1    | 2    | **21**          | **49**          | **789**         | 2              | 19             | 1092           |
> |                  | 10    | 2    | 1      | 1  | 3-4    | **18**          | **44**          | **1321**        | 1              | 7              | 1549           |

---

> > ### Comment · Reviewer_fzvQ · 2024-11-26
> > **Official Comment by Reviewer fzvQ**
> >
> > Thank you for your clarifications and comments.
> >
> > After reading the rest of the reviews and comments,
> > * I agree with Reviewer BB3F about adding the details of the actual setup, it will enhance the paper's readability.
> > * I agree with Reviewer Jcan about the clarity of motivation for MultiRoomR.
> >
> > Please revise the submission with what you provided in the responses.
> >
> > > We provide results for the flat object-oriented POMDP here(ablation on hierarchy).....
> >
> > Thank you for providing these results. Now, it is clear that the proposed hierarchical planning is essential. Thus, I raise my score to 6 instead of 3.
> >
> > **Why not higher?**
> >
> > I still believe that the novelty is limited.
> > > Extend OO-POMDP originally designed for object search to rearrangement tasks.
> >
> > Despite explaining the extended problem formulation, I think it is not enough for a higher score. I have raised the score due to the shown increase in performance using hierarchical planning, which aligns with the second novelty mentioned.
> >
> > Furthermore, it is hard to evaluate the performance of this work without providing any direct comparison to other baselines (also commented by other reviewers). I fully understand the constraints mentioned in the responses (unavailability of codes and different problem settings), so I do not consider this drawback. However, I believe revising the submission with the reasons provided in the responses is important.

---

### Author Response · Authors · 2024-11-17
**Common response**

Common Response:

We thank the reviewers for their thoughtful feedback. We are encouraged they found our work to be novel in addressing the complex multi-room rearrangement problem (R2, R4), with significant contributions through our HOO-POMDP framework, achieving comparable results to perfect knowledge baselines (R1), and effectively handling uncertainties in large multi-room spaces (R4). We appreciate the recognition of our technical contributions in handling imperfect detection and low-level control (R2), and addressing the challenging blocked path scenarios (R3). We are pleased that reviewers found our new Multi RoomR dataset to be valuable (R1, R2, R4), and our supplementary video to be informative (R4). We are glad they recognized our work's importance for long-term generalist robots (R3) and its significance to the field (R2). We address reviewer comments below and will incorporate all feedback.

**Proposed MultiRoomR Dataset details**

1. Size of dataset: 300 room configurations, ten rearrangements each
2. Types of objects selected: Present in the appendix section A.1.2 (table 2)
3. The rationale behind selecting them: Almost all object types in AI2Thor are selected. Objects that are too small are removed, as they are undetectable even from a close distance.
4. Room size information:\
    a. 200 room configurations of 2 rooms. (50% contain blocked path)
    b. 50 room configurations of  3 rooms. (100% contain blocked paths)
    c. 50 room configurations of 4 rooms. (100% contain blocked paths).

5. Criteria for object placement:\
    a. Criteria 1: At least one object needs to be moved in every room This ensures that the agent must explore all rooms to complete the task.
    b. Criteria 2: For blocked goal and swap cases: We generate scenes where one object blocks the goal location of another object or two objects block each other’s goal (swap).
    c. Criteria 3: For blocked path scenes, the location of the object blocking the path is chosen to maximize the area of the house that is inaccessible.

We will add all these dataset details in the appendix.

---

### Meta-Review · Area_Chair_i5p7 · 2024-12-22

**Metareview:**

The submission presents a method for solving multi-room object rearrangement problems based on planning in a hierarchical object-oriented POMDP and evaluates in the AI2-THOR environment. It also introduces MultiRoomR, a new set of multi-room object rearrangement task instances.

According to the reviews, the paper has the following strengths:

- The proposed method works well, despite partial object observability and without assuming perfect navigation, motion planning or manipulation, on many challenging task instances. This includes instances from the newly introduced MultiRoomR set, which are generally harder than the existing ones.

- The new MultiRoomR set is a welcome contribution to the research community.


The weaknesses identified by the reviewers are:

- Arguable novelty. Methodologically, the paper's approach is an extension of object-oriented POMDPs from object search tasks to rearrangement in multi-room environments.

- Clarity and (lack of) comparisons to the existing methods. These two got partly addressed during the discussion.

The metareviewer finds that in addition to the submission's pros and cons surfaced in the reviews and the ensuing discussion, an important additional consideration in this paper's case is the issue of its contributions' scope. The proposed method and dataset are structurally engineered to solve a very specific class of embodied AI tasks, multi-room object rearrangement under partial observability. It is unclear which of the proposed method's aspects can be applied to other tasks, let alone how to extend the entire method to them. But this class of tasks is entirely synthetic. The research community uses its instances as benchmarks for evaluating embodied AI systems' planning and reasoning capabilities and focuses on the instances artificially designed to be hard from the planning and reasoning standpoint. These tasks were originally inspired by real-life object rearrangement but are very far from the rearrangement instances an embodied AI agent is likely to encounter in the real world. This gap is fine if the intent is to use these tasks for the usual purpose of a benchmark, i.e., comparing the performance of different methods, but is problematic from the standpoint of crafting a method for solving this specific benchmark, since it's difficult to make a claim that such a method solves, or will ever be able to solve, a real problem. The paper's approach unintentionally illustrates this issue: separating rearrangement in two distinct stages of information gathering and object manipulation is OK when solving an artificial benchmark but would be very unnatural in reality. It is also hard to imagine that, if faced with a rearrangement problem, an embodied AI agent would switch to using this highly specialized solver rather than a more general reasoning module.

These considerations make the metareviewer recommend rejection, since the paper's contribution is essentially engineered to solve and extend a benchmark and doesn't seem to generalize beyond that benchmark. This contribution can still be of interest at a venue that focuses specifically on benchmarks like this, but its scope is too narrow for ICLR.

**Additional Comments On Reviewer Discussion:**

The discussion has helped addresses some concerns around clarity and evaluation. With regards to novelty, the discussion has surfaced useful details but overall confirmed the reviewers' original opinions that the methodological contribution is incremental.

---

### Decision · Program_Chairs · 2025-01-22

Reject